# TRANSITIVE RL:
# VALUE LEARNING VIA DIVIDE AND CONQUER

**Seohong Park**[*1]   **Aditya Oberai**[*1]   **Pranav Atreya**[1]   **Sergey Levine**[1]

[1]University of California, Berkeley

{seohong, aoberai}@berkeley.edu

## ABSTRACT

In this work, we present Transitive Reinforcement Learning (TRL), a new value learning algorithm based on a divide-and-conquer paradigm. TRL is designed for offline goal-conditioned reinforcement learning (GCRL) problems, where the aim is to find a policy that can reach any state from any other state in the smallest number of steps. TRL converts a triangle inequality structure present in GCRL into a practical divide-and-conquer value update rule. This has several advantages compared to alternative value learning paradigms. Compared to temporal difference (TD) methods, TRL suffers less from bias accumulation, as in principle it only requires $O(\log T)$ recursions (as opposed to $O(T)$ in TD learning) to handle a length-$T$ trajectory. Unlike Monte Carlo methods, TRL suffers less from high variance as it performs dynamic programming. Experimentally, we show that TRL achieves the best performance in highly challenging, long-horizon benchmark tasks compared to previous offline GCRL algorithms.

Blog post: https://seohong.me/blog/rl-without-td-learning

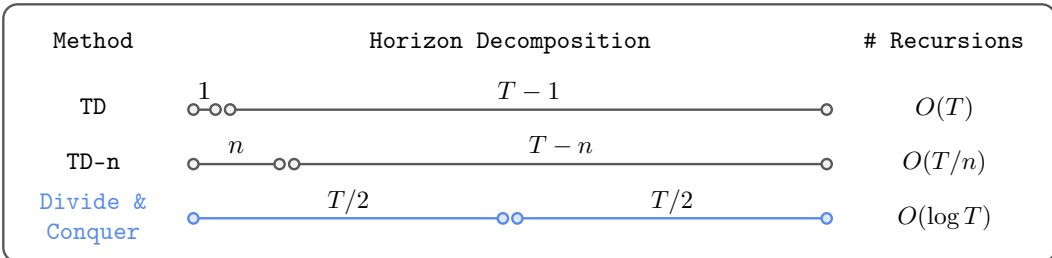

| Method | Horizon Decomposition | # Recursions |
|---|---|---|
| TD | 1 — $T-1$ | $O(T)$ |
| TD-n | $n$ — $T-n$ | $O(T/n)$ |
| Divide & Conquer | $T/2$ — $T/2$ | $O(\log T)$ |

Figure 1: **Transitive RL.** Transitive RL is based on the divide-and-conquer paradigm, which can in theory reduce the number of Bellman recursions to $O(\log T)$ in the best case, unlike TD-based methods.

## 1 INTRODUCTION

A fundamental challenge in off-policy reinforcement learning (RL) is the curse of horizon (Liu et al., 2018). In off-policy RL, we typically train a value function using (some variant of) temporal difference (TD) learning with Bellman backups, $Q(s, a) \leftarrow r(s, a) + \gamma \max_{a'} \bar{Q}(s', a')$, and train a policy to maximize the learned value function. The problem is that, each Bellman update involves regression toward a *biased* target value, and these biases *accumulate* over the horizon through recursion. This bias accumulation is one of the main obstacles hindering the scaling of off-policy RL to complex, long-horizon tasks (Park et al., 2025b).

A standard technique to mitigate this bias accumulation issue in practice is to mix Monte Carlo (MC) returns, either for short-horizon trajectory chunks (*e.g.*, $n$-step returns) or even for full trajectories (*e.g.*, purely MC-based approaches). While these approaches do mitigate bias accumulation, they now suffer from *higher variance* as the Monte Carlo horizon grows. Sometimes, it is possible to strike a balance in the bias-variance tradeoff using $n$-step returns with an appropriately tuned

---
[*]Equal contribution.

$n$ (Sutton & Barto, 2005; Park et al., 2025b). However, this does not *fundamentally* solve the curse of horizon, as it reduces the horizon length only by a constant factor ($n$). We also need to carefully tune $n$ for each task, and it requires optimality in length-$n$ trajectory chunks.

In this work, we investigate whether we can find a different mechanism to mitigate the curse of horizon in an important subset of RL problems: offline goal-conditioned RL (GCRL) (Kaelbling, 1993; Park et al., 2025a). Goal-conditioned RL aims to find a policy that can reach any state from any other state in the smallest number of steps. This is typically done by learning the optimal distance function $d^*(s, g)$ (which corresponds to the value function), defined as the minimal number of steps to reach $g$ from $s$ (see Section 3 for the formal definition).

By its shortest-path-like nature, the optimal distance function $d^*$ satisfies the following triangle inequality in deterministic environments:

$$d^*(s, g) \leq d^*(s, w) + d^*(w, g) \tag{1}$$

for all states $s, w, g$. This provides an additional **divide-and-conquer** structure that we can exploit. That is, we can compute the distance $d^*(s, g)$ by combining information about two "smaller" distances, $d^*(s, w)$ and $d^*(w, g)$.

Our central hypothesis in this work is that, a *divide-and-conquer* value learning algorithm can mitigate the curse of horizon more effectively than standard TD- or MC-based methods in offline goal-conditioned RL. Unlike TD-based methods, divide and conquer has the potential to *logarithmically* reduce the number of biased recursions in principle, thanks to its recursive nature (Figure 1). Unlike MC-based methods, divide and conquer suffers less from the high variance of long-horizon MC returns, as it performs dynamic programming.

Based on this idea, we develop a new divide-and-conquer-based offline goal-conditioned RL algorithm, **Transitive RL (TRL)**, for long-horizon goal-reaching tasks. There are several scaling challenges in implementing this idea in practice, such as value overestimation. We address these challenges with newly introduced ideas, such as in-trajectory subgoals and distance re-weighting.

Our main contributions are twofold. First, we propose TRL, a new scalable, practical value learning algorithm for offline goal-conditioned RL based on divide and conquer. Second, we empirically demonstrate TRL often exhibits better performance than previous TD- and MC-based approaches on highly complex, long-horizon robotics tasks over 1000 environment steps. To the best of our knowledge, TRL is the first divide-and-conquer value learning method that scales to long-horizon robotic tasks beyond toy domains.

## 2 RELATED WORK

**Offline RL and offline goal-conditioned RL.** Much work has studied the offline reinforcement learning problem, which aims to find the best return-maximizing policy within the support of the offline data. Most works distinguish themselves from the way they mitigate value overestimation for out-of-distribution actions, usually either imposing conservatism constraints on the value function (Kumar et al., 2020), behavior regularizing the actor (Wu et al., 2019; Fujimoto & Gu, 2021; Tarasov et al., 2023; Gao et al., 2025; Park et al., 2025c), or implicitly performing Bellman updates without querying out-of-distribution actions (Kostrikov et al., 2022; Xu et al., 2023; Garg et al., 2023). In offline goal-conditioned RL, whose aim is to train a multi-task policy that can reach any state, previous methods either extend offline RL algorithms to the goal-conditioned setting (Yang et al., 2022; Park et al., 2023) or leverage specific structures of the goal-conditioned RL problem with quasimetric learning (Wang et al., 2023), hierarchical policy extraction (Park et al., 2023), or probabilistic interpretation (Eysenbach et al., 2021; 2022; Zheng et al., 2024). In this work, we propose a new goal-conditioned RL algorithm based on a triangle inequality, as discussed in the next paragraph.

**The triangle inequality in GCRL.** Our main idea in this work is based on the triangle inequality in goal-conditioned RL (Equation (7)). The idea of using the triangle inequality dates back to the very first work in goal-conditioned RL by Kaelbling (1993). Since then, several prior works have leveraged this structure in three main ways. "Hard" approaches strictly impose the triangle inequality via an architectural quasimetric constraint (Wang et al., 2023; Myers et al., 2024), "soft" approaches use the triangle inequality as a value backup (Kaelbling, 1993; Dhiman et al., 2018; Jur-

genson et al., 2020; Piekos et al., 2023), and planning-based approaches employ this inequality for planning (Eysenbach et al., 2019; Parascandolo et al., 2020; Jurgenson et al., 2020; Li et al., 2021). Among planning-free off-policy RL methods, while hard approaches have proven effective in practice so far (*e.g.*, QRL (Wang et al., 2023)), soft approaches (to our knowledge) have not yet scaled beyond 2-D toy environments in the absence of additional planning (Kaelbling, 1993; Dhiman et al., 2018; Jurgenson et al., 2020; Piekos et al., 2023).

Our method, TRL, is based on the soft triangle inequality; *i.e.*, we use the triangle inequality for value backups, instead of hard quasimetric constraints. However, unlike previous soft approaches, we show that TRL scales to highly complex, long-horizon robotic tasks with 1B-sized datasets, even outperforming hard (*i.e.*, quasimetric-based) approaches. This is made possible by our key techniques (in-sample maximization within in-sample subgoals; see Section 4.2), which prevent value overestimation when applying the triangle inequality in practice.

# 3 PRELIMINARIES

**Problem setting.** We consider a controlled Markov process defined by a tuple $\mathcal{M} = (\mathcal{S}, \mathcal{A}, p)$, where $\mathcal{S}$ is the state space, $\mathcal{A}$ is the action space, and $p(s' \mid s, a) : \mathcal{S} \times \mathcal{A} \to \Delta(\mathcal{S})$ is the transition dynamics distribution. In the definition above, $\Delta(\mathcal{X})$ denotes the set of probability distributions on a space $\mathcal{X}$ and placeholder variables are denoted in gray. We assume that we are given an unlabeled dataset $\mathcal{D} = \{\tau^{(i)}\}_i$ consisting of length-$T$ reward-free trajectories $\tau = (s_0, a_0, s_1, \ldots, s_T)$ (in practice, different trajectories can have different lengths, but we assume they have the same length for the simplicity of discussion).

In this work, we aim to develop a better recipe for offline goal-conditioned RL. Offline GCRL aims to learn a goal-conditioned policy $\pi(a \mid s, g) : \mathcal{S} \times \mathcal{S} \to \mathcal{A}$ that maximizes the objective $\tilde{V}^\pi(s, g) = \mathbb{E}_{\tau \sim p^\pi(\tau \mid s, g)}[\sum_{t=0}^{T} \gamma^t \mathbb{I}(s_t = g)]$ for all $s, g \in \mathcal{S}$ purely from the dataset $\mathcal{D}$ (without additional environment interactions), where $\mathbb{I}(\cdot)$ denotes the 0-1 indicator function,[1] $\gamma \in (0, 1)$ denotes the discount factor, and $p^\pi(\tau \mid s, g) = \pi(a_0 \mid s_0, g) p(s_1 \mid s_0, a_0) \cdots p(s_T \mid s_{T-1}, a_{T-1})$ where $s_0 = s$. Here, we note that states and goals live in the same space $\mathcal{S}$, and the reward function is simply given as the 0-1 sparse reward function. We also consider a variant of the above goal-conditioned RL objective defined by hitting times, where the agent enters an absorbing state upon reaching the goal (*i.e.*, gets a reward of 1 at most once) (Wang et al., 2023). We denote the corresponding hitting-time-based value function as $V^\pi(s, g)$. In this work, we mainly consider this hitting-time variant, as it has a close connection to temporal distances, as discussed in the next paragraph.

As commonly done by many previous works in goal-conditioned RL (Eysenbach et al., 2019; Janner et al., 2022; Ghosh et al., 2023; Park et al., 2023; Wang et al., 2023; Park et al., 2024b), we assume that the environment dynamics are deterministic, unless mentioned otherwise. In deterministic environments, we can define the notion of temporal distances. The temporal distance $d^*(s, g)$ from $s \in \mathcal{S}$ to $g \in \mathcal{S}$ is defined as the minimum number of steps to reach $g$ from $s$ ($\infty$ if it is not reachable). From the definition, it is straightforward to see that $V^*(s, g) := \max_\pi V^\pi(s, g) = \gamma^{d^*(s, g)}$ in the deterministic case.

## 3.1 PREVIOUS APPROACHES TO OFFLINE GCRL

An offline RL algorithm typically consists of two components: value estimation and policy extraction. In this section, we review two representative paradigms for goal-conditioned value learning (MC and TD), and two standard techniques for policy extraction.

**Monte Carlo (MC) value estimation.** Monte Carlo value estimation is one of the simplest techniques for goal-conditioned value learning. It trains a behavioral value function based on the distances achieved in dataset trajectories. Among MC-based approaches, distance regression methods (Tian et al., 2021; Shah et al., 2021) fit a Q function $Q(s, a, g) : \mathcal{S} \times \mathcal{A} \times \mathcal{S} \to \mathbb{R}$ with the fol-

---

[1]We assume that the state space is discrete for notational simplicity. This is only to simplify mathematical notation, and most discussions in this paper can be directly extended to continuous state spaces as well with appropriate measure-theoretic formulations (Blier et al., 2021).

lowing loss:

$$L^{\mathrm{MC}}(Q) = \mathbb{E}_{\substack{\tau \sim \mathcal{D}, \\ 0 \leq i \leq j < T}} \left[ (Q(s_i, a_i, s_j) - \gamma^{j-i})^2 \right], \qquad (2)$$

where $\tau$ is sampled uniformly from the dataset, and $i$ and $j$ are sampled uniformly from the set $\{0, 1, \ldots, T-1\}$. While this objective is biased when either the dynamics or the data-collecting policy is stochastic (Akella et al., 2023), it is simple, stable, and often performs surprisingly well in practice (Tian et al., 2021; Shah et al., 2021; Hejna et al., 2023).

**Temporal difference (TD) value learning.** Another class of methods employs temporal difference learning to learn a goal-conditioned value function. As an example, goal-conditioned implicit Q-learning (GCIQL) (Kostrikov et al., 2022; Park et al., 2023) fits value functions with the following losses:

$$L^{\mathrm{IQL-V}}(V) = \mathbb{E}_{s,a,g \sim \mathcal{D}} \left[ \ell_\kappa^2(V(s,g) - \bar{Q}(s,a,g)) \right], \qquad (3)$$

$$L^{\mathrm{IQL-Q}}(Q) = \mathbb{E}_{s,a,s',g \sim \mathcal{D}} \left[ (Q(s,a,g) - \mathbb{I}(s=g) - \gamma V(s',g))^2 \right], \qquad (4)$$

where goals are typically sampled with hindsight relabeling (Andrychowicz et al., 2017; Park et al., 2025a), $\ell_\kappa^2$ denotes the expectile loss with a parameter $\kappa \in [0.5, 1)$, $\ell_\kappa^2(x) = |\kappa - \mathbb{I}(x < 0)|x^2$, and $\bar{Q}$ denotes the target Q function (Mnih et al., 2013). The asymmetric expectile regression in Equation (3) approximates the max operator in the Bellman update ($V(s,g) \leftarrow \max_{a \in \mathcal{A}} Q(s,a,g)$) only with in-sample actions, avoiding querying the Q function with out-of-distribution actions. We note that when $\kappa = 0.5$, the above objective yields a behavioral (SARSA) goal-conditioned value function (Brandfonbrener et al., 2021; Kostrikov et al., 2022; Park et al., 2025b).

**Policy extraction.** After learning a goal-conditioned value function, the next step is to train a policy to maximize the learned values. This procedure is called policy extraction, and we describe two standard techniques below.

Reparameterized gradients (Fujimoto & Gu, 2021) extract a Gaussian policy by maximizing the following "DDPG+BC" objective (Park et al., 2024a):

$$J^{\mathrm{DDPG+BC}}(\pi) = \mathbb{E}_{\substack{s,a,g \sim \mathcal{D}, \\ a^\pi \sim \pi(a|s,g)}} \left[ Q(s, a^\pi, g) + \alpha \log \pi(a \mid s, g) \right], \qquad (5)$$

where $a^\pi$ is a reparameterized sample from the policy and $\alpha$ is the strength of the BC constraint. Intuitively, this maximizes the learned value function while not deviating too much from the dataset distribution to prevent out-of-distribution exploitation.

Rejection sampling (Ghasemipour et al., 2021; Chen et al., 2023; Hansen-Estruch et al., 2023) simply defines a policy to be the $\arg\max$ of behavioral action samples that maximize the value function:

$$\pi(s,g) \overset{d}{=} \underset{a_1, \cdots, a_N : a_i \sim \pi^\beta(a|s,g)}{\arg\max} Q(s, a_i, g), \qquad (6)$$

where $N$ denotes the number of samples, $\overset{d}{=}$ denotes equality in distribution, and $\pi^\beta$ denotes a goal-conditioned BC policy. Typically, the BC policy is modeled by an expressive generative model (*e.g.*, diffusion models (Sohl-Dickstein et al., 2015; Ho et al., 2020) and flow matching (Lipman et al., 2024)) so that it can effectively capture the potentially multi-modal distributions of the behavioral policy (Chen et al., 2023; Hansen-Estruch et al., 2023; Park et al., 2025b).

## 4 TRANSITIVE RL

As motivated in Section 1, our main aim is to develop a practical *divide-and-conquer* value learning algorithm for offline goal-conditioned RL. Offline goal-conditioned RL provides a natural shortest-path structure that is amenable to divide and conquer. Namely, in deterministic environments,[2] the temporal distance function $d^*(s, g)$ always satisfies the following triangle inequality (Kaelbling, 1993; Dhiman et al., 2018; Wang et al., 2023; Piekos et al., 2023):

$$d^*(s, g) \leq d^*(s, w) + d^*(w, g) \qquad (7)$$

---

[2]While we mainly consider deterministic environments in this work, we note that the GCRL problem exhibits a similar triangle-inequality structure in stochastic environments as well; see Myers et al. (2024).

for all $s, w, g \in \mathcal{S}$. The equality holds when $w$ is on a shortest path from $s$ to $g$. Throughout this paper, we call $w$ a *subgoal*.

Equivalently, from the relation $V^*(s, g) = \gamma^{d^*(s,g)}$, we can rewrite this in terms of the optimal value function $V^*(s, g)$ as follows:

$$V^*(s, g) \geq V^*(s, w)V^*(w, g). \tag{8}$$

This motivates the following *transitive* Bellman update rule for a value function $V(s, g)$:

$$V(s, g) \leftarrow \begin{cases} \gamma^0 & \text{if } s = g, \\ \gamma^1 & \text{if } (s, g) \in \mathcal{E}, \\ \max_{w \in \mathcal{S}} V(s, w)V(w, g) & \text{otherwise,} \end{cases} \tag{9}$$

where $\mathcal{E}$ (the edge set) denotes the set of $(s, s') \in \mathcal{S} \times \mathcal{S}$ pairs such that $s \neq s'$ and $s'$ is reachable from $s$ by a single action. If we initialize the value function with $V(s, g) \leq 0$ for all $s, g \in \mathcal{S}$, a standard result in computer science (*e.g.*, the Floyd-Warshall algorithm) implies that repeatedly applying this operator converges to the optimal value function $V^*(s, g)$ (Dhiman et al., 2018; Piekos et al., 2023).

We also have a variant of the above update rule for an action-value function $Q(s, a, g)$:

$$Q(s, a, g) \leftarrow \begin{cases} \gamma^0 & \text{if } s = g, \\ \gamma^1 & \text{if } g = p(s, a) \text{ and } s \neq g, \\ \max_{w \in \mathcal{S}, a' \in \mathcal{A}} Q(s, a, w)Q(w, a', g) & \text{otherwise,} \end{cases} \tag{10}$$

where we slightly abuse the notation to denote $p$ as the deterministic dynamics function.

**Why do we want to use the triangle inequality?** As described in Section 1, the main benefit of the triangle inequality is that we can perform divide and conquer instead of TD or MC value learning. Unlike TD learning, which requires $O(T)$ Bellman recursions, divide and conquer can reduce the number of Bellman recursions to $O(\log T)$ in the best case (in theory, with optimally chosen subgoals). Unlike MC learning, which suffers from high variance with long horizons, divide and conquer performs dynamic programming (*i.e.*, aggregates intermediate values) and thus mitigates the variance issue. Hence, we expect that the use of the triangle inequality potentially leads to a scalable value learning algorithm that scales better in *long-horizon* tasks.

## 4.1 THE CHALLENGE

While the transitive Bellman update rule (Equation (9)) provides a natural way to perform divide and conquer for goal-conditioned value learning, directly implementing this idea in practice is not straightforward. The main problem is the $\max_{w \in \mathcal{S}}$ operator in Equation (9). While we can simply iterate over the set of states to compute the maximum in the tabular case, doing so would lead to (potentially catastrophic) value overestimation under function approximation, because the max operator over a large number of subgoals can easily exploit any single subgoal's positively biased value estimation error. This issue is further exacerbated in the offline setting, where value overestimation poses a particularly severe challenge (Levine et al., 2020).

To address this challenge, previous works have mainly considered two practical solutions. Jurgenson et al. (2020) use a previous iteration of the value network (similar to target networks (Mnih et al., 2013)) for the target value in Equation (9). Piekos et al. (2023) train a separate generator network $G(s, a, g) : \mathcal{S} \times \mathcal{A} \times \mathcal{S} \rightarrow \mathcal{S}$ to approximate the $\max_{w \in \mathcal{S}}$ operator, similar to DDPG (Lillicrap et al., 2016). However, as we will show in our experiments, these techniques are not sufficient to stabilize training, leading to zero success rates in most simulated robotic benchmark tasks. Indeed, most prior works that use the transitive Bellman update without explicit planning have demonstrated their methods only on 2-D toy tasks, such as point mazes and discrete grid worlds (Kaelbling, 1993; Dhiman et al., 2018; Jurgenson et al., 2020; Piekos et al., 2023).

## 4.2 THE IDEA

Our key idea to handle the overestimation issue in the $\max$ operator is to perform *in-sample maximization* using only *in-trajectory subgoals*. Specifically, we apply the following two modifications.[3]

First, we replace the strict $\max$ operator in Equation (9) with soft expectile regression (Newey & Powell, 1987; Kostrikov et al., 2022). That is, we (conceptually) minimize the following loss:

$$\mathbb{E}[\ell_\kappa^2(V(s,g) - \bar{V}(s,w)\bar{V}(w,g))], \tag{11}$$

where $\ell_\kappa^2$ denotes the expectile loss described in Section 3, and $\bar{V}$ denotes the target network (Mnih et al., 2013). This expectile loss allows us to approximate the maximum in Equation (9) without having to iterate over the states explicitly.

Second, we only consider *in-trajectory* states as potential subgoals (we call them *behavioral subgoals*). That is, for a dataset trajectory $\tau = (s_0, a_0, s_1, \ldots, s_T)$, we update $V(s_i, s_j)$ only using $V(s_i, s_k)V(s_k, s_j)$ with $i \leq k \leq j$, instead of considering arbitrary states as subgoals. While this decision might seem restrictive at first glance, we find that restricting to behavioral subgoals is crucial to make the transitive Bellman update work in practice (Section 5.3). Otherwise, the probability of an arbitrary state being a "valid" subgoal is low, and thus we would need much more aggressive maximization (*i.e.*, a higher expectile in Equation (11)), which leads to unstable training. Moreover, we experimentally find that behavioral subgoals can still be highly effective even when the dataset consists of uniformly random atomic motions. This is analogous to how behavioral value functions (*i.e.*, one-step RL) are often sufficient in practice (Brandfonbrener et al., 2021; Eysenbach et al., 2022; Park et al., 2025b) even on suboptimal datasets.

## 4.3 PRACTICAL RECIPE

Based on the high-level idea in Section 4.2, we now describe the full recipe of our method, which we call **Transitive RL (TRL)**. TRL trains a goal-conditioned action-value function $Q(s, a, g) : \mathcal{S} \times \mathcal{A} \times \mathcal{S} \to \mathbb{R}$ based on the Q-based transitive Bellman operator (Equation (10)), and then extracts a goal-conditioned policy $\pi(a \mid s, g) : \mathcal{S} \times \mathcal{S} \to \Delta(\mathcal{A})$.

### 4.3.1 VALUE LEARNING

TRL's value learning is based on the high-level idea described in Section 4.2. In practice, we additionally employ the following technique to further improve performance and stability.

**Distance-based re-weighting.** In transitive Bellman updates, the accuracy of the target value for a longer trajectory chunk ($s_i$ to $s_j$) depends on those of two shorter trajectory chunks ($s_i$ to $s_k$ and $s_k$ to $s_j$). Hence, it is particularly important to keep the shorter trajectory chunks' values accurate. To this end, we weigh the loss for each sample $(s_i, s_j)$ in the batch by $w(s_i, s_j) := 1/(1 + \log_\gamma Q(s_i, a_i, s_j))^\lambda$, where $\lambda$ is a hyperparameter. This adjusts the weight for each chunk to be (roughly) inversely proportional to its estimated distance, thus focusing more on shorter trajectory chunks. This bears similarity to classical dynamic programming where smaller subproblems are solved before larger ones.

With this technique, the final value loss for TRL is defined as follows:

$$L^{\mathrm{TRL}}(Q) = \mathbb{E}_{\tau \sim \mathcal{D}}\left[w(s_i, s_j)D_\kappa\left(Q(s_i, a_i, s_j), \bar{Q}(s_i, a_i, s_k)\bar{Q}(s_k, a_k, s_j)\right)\right]. \tag{12}$$

where $D$ denotes a loss function (*e.g.*, squared regression, binary cross-entropy, etc.), and $D_\kappa$ denotes its expectile variant, $D_\kappa(x, y) := |\kappa - \mathbb{I}(x > y)|D(x, y)$. In our experiments, we use the binary cross-entropy (BCE) loss, following prior work in goal-conditioned RL (Kalashnikov et al., 2018; Eysenbach et al., 2022; Park et al., 2025b). Also, to handle the base cases, we replace $\bar{Q}(s_i, a_i, s_k)$ with $\gamma^{k-i}$ if $k - i \leq 1$ and $\bar{Q}(s_k, a_k, s_j)$ with $\gamma^{j-k}$ if $j - k \leq 1$ in the above loss.

### 4.3.2 POLICY EXTRACTION

After training a goal-conditioned value function, we extract a policy to maximize the learned value. In TRL, we consider two policy extraction methods, reparameterized gradients and rejection sam-

---

[3]In this section, for notational simplicity, we describe our high-level idea based on the V version of the transitive Bellman operator (Equation (9)). In practice, we use the Q version (Equation (10)).

> **Digression:** Does random midpoint sampling still lead to logarithmic Bellman recursions?
>
> While we motivated divide-and-conquer value learning from its logarithmic dependency on the horizon in Section 1, TRL does not always divide a trajectory into two equal-sized chunks. It instead *samples* a random subgoal $s_k$ between $s_i$ and $s_j$. One might wonder whether TRL's random "sampling" variant still gives us $O(\log T)$ recursions. The answer is still yes (in the ideal, tabular case), and we provide a proof in Section A.

---

**Algorithm 1** Transitive Reinforcement Learning (TRL)

---

▷ Value learning
Initialize value function $Q(s, a, g)$, policy $\pi(a \mid s, g)$
**while** not converged **do**
    Sample $\tau = (s_0, a_0, s_1, \ldots, s_T) \sim \mathcal{D}$
    Sample $i, j \sim \text{Unif}(\{0, 1, \ldots, T-1\})$ such that $i < j$
    Sample $k \sim \text{Unif}(\{i, i+1, \ldots, j-1\})$
    Train $Q$ by minimizing $L^{\text{TRL}}(Q)$ (Equation (12))

▷ Policy extraction (can be run in parallel with above)
Extract $\pi$ using either reparameterized gradients (Equation (5)) or rejection sampling (Equation (6))

---

pling (Section 3). By default, we use reparameterized gradients, but we find rejection sampling to work better on long-horizon puzzle tasks, where the behavioral policy is highly multi-modal.

We provide pseudocode for TRL in Algorithm 1.

## 5 EXPERIMENTS

In this section, we empirically answer several research questions about divide-and-conquer value learning and TRL. In Section 5.1, we compare TRL's divide-and-conquer update rule with previous TD- and MC-based update rules on large-scale, long-horizon tasks. In Section 5.2, we compare TRL with previous offline GCRL methods on a standard benchmark suite. In Section 5.3, we provide various ablation studies of TRL. We use four random seeds unless otherwise stated, and present 95% confidence intervals in plots and standard deviations in tables. In tables, we highlight numbers that are at or above 95% of the best performance in blue, as in Park et al. (2025a).

### 5.1 IS DIVIDE AND CONQUER BETTER THAN TD AND MC ON LONG-HORIZON TASKS?

Our first goal is to see whether TRL's divide-and-conquer value learning algorithm exhibits better scalability to **long-horizon** problems than more standard approaches (TD, $n$-step TD, and MC) in practice. This was our main motivation for TRL (Section 1).

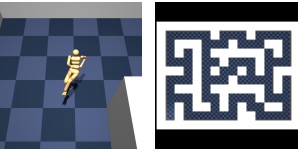

humanoidmaze-giant

**Tasks.** To empirically answer this question, we employ three highly complex, long-horizon environments used in a recent horizon scaling study by Park et al. (2025b): humanoidmaze-giant, puzzle-4x5, and puzzle-4x6 (Figure 2).[4] In humanoidmaze-giant, the agent must control a humanoid robot with 21 degrees of freedom to navigate a large maze. In puzzle-4x5 and puzzle-4x6, the agent must control a robot arm to press buttons to solve the "Lights Out" puzzle. Each environment provides five tasks (*i.e.*, five state-goal pairs) for evaluation. These tasks are highly challenging and have long horizons: for example, the test-time task horizon of humanoidmaze-giant is 4000. For datasets, we employ the 1B-sized ones used by Park et al. (2025b).

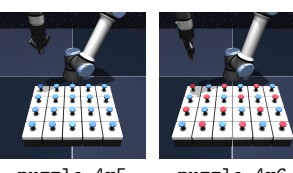

puzzle-4x5     puzzle-4x6

Figure 2: **Long-horizon tasks.**

---

[4]We omit cube-octuple from Park et al. (2025b), as none of the methods achieve non-trivial performance on this task without hierarchical *policies*, which are orthogonal to the focus of this paper (*i.e.*, value learning).

Table 1: **Results on large-scale, *long-horizon* tasks.** TRL achieves the best performance on highly challenging, long-horizon benchmark tasks that require up to 3000 environment steps.

| Environment | Task | FBC | IQL | CRL | SAC+BC | WGCSL | HIQL | QRL | TDP | COE | TD | MC | TRL |
|---|---|---|---|---|---|---|---|---|---|---|---|---|---|
| humanoidmaze-giant | task1 | $0_{\pm0}$ | $0_{\pm0}$ | $52_{\pm38}$ | $5_{\pm3}$ | $0_{\pm0}$ | $13_{\pm14}$ | $1_{\pm1}$ | $0_{\pm0}$ | $1_{\pm1}$ | $2_{\pm4}$ | $64_{\pm9}$ | $71_{\pm15}$ |
| | task2 | $0_{\pm0}$ | $3_{\pm7}$ | $63_{\pm46}$ | $12_{\pm3}$ | $1_{\pm1}$ | $32_{\pm18}$ | $2_{\pm2}$ | $3_{\pm3}$ | $2_{\pm2}$ | $8_{\pm4}$ | $87_{\pm5}$ | $87_{\pm6}$ |
| | task3 | $0_{\pm0}$ | $5_{\pm6}$ | $68_{\pm46}$ | $8_{\pm6}$ | $1_{\pm1}$ | $23_{\pm18}$ | $2_{\pm3}$ | $3_{\pm2}$ | $4_{\pm2}$ | $1_{\pm1}$ | $83_{\pm8}$ | $44_{\pm8}$ |
| | task4 | $0_{\pm0}$ | $2_{\pm3}$ | $57_{\pm41}$ | $2_{\pm3}$ | $1_{\pm1}$ | $10_{\pm4}$ | $0_{\pm0}$ | $0_{\pm0}$ | $1_{\pm2}$ | $2_{\pm2}$ | $78_{\pm8}$ | $94_{\pm4}$ |
| | task5 | $0_{\pm0}$ | $3_{\pm4}$ | $68_{\pm46}$ | $0_{\pm0}$ | $2_{\pm3}$ | $35_{\pm14}$ | $8_{\pm1}$ | $4_{\pm2}$ | $5_{\pm2}$ | $3_{\pm1}$ | $84_{\pm11}$ | $99_{\pm1}$ |
| | overall | $0_{\pm0}$ | $3_{\pm4}$ | $62_{\pm42}$ | $5_{\pm0}$ | $1_{\pm1}$ | $23_{\pm13}$ | $3_{\pm2}$ | $2_{\pm0}$ | $2_{\pm1}$ | $3_{\pm2}$ | $79_{\pm4}$ | $79_{\pm2}$ |
| puzzle-4x5 | task1 | $0_{\pm0}$ | $100_{\pm0}$ | $7_{\pm5}$ | $95_{\pm3}$ | $0_{\pm0}$ | $38_{\pm26}$ | $0_{\pm0}$ | $0_{\pm0}$ | $0_{\pm0}$ | $84_{\pm7}$ | $100_{\pm0}$ | $100_{\pm0}$ |
| | task2 | $0_{\pm0}$ | $0_{\pm0}$ | $0_{\pm0}$ | $0_{\pm0}$ | $0_{\pm0}$ | $0_{\pm0}$ | $0_{\pm0}$ | $0_{\pm0}$ | $0_{\pm0}$ | $4_{\pm3}$ | $67_{\pm12}$ | $99_{\pm1}$ |
| | task3 | $0_{\pm0}$ | $0_{\pm0}$ | $0_{\pm0}$ | $0_{\pm0}$ | $0_{\pm0}$ | $0_{\pm0}$ | $0_{\pm0}$ | $0_{\pm0}$ | $0_{\pm0}$ | $2_{\pm2}$ | $8_{\pm10}$ | $100_{\pm0}$ |
| | task4 | $0_{\pm0}$ | $0_{\pm0}$ | $0_{\pm0}$ | $0_{\pm0}$ | $0_{\pm0}$ | $0_{\pm0}$ | $0_{\pm0}$ | $0_{\pm0}$ | $0_{\pm0}$ | $2_{\pm2}$ | $50_{\pm9}$ | $99_{\pm1}$ |
| | task5 | $0_{\pm0}$ | $0_{\pm0}$ | $0_{\pm0}$ | $0_{\pm0}$ | $0_{\pm0}$ | $0_{\pm0}$ | $0_{\pm0}$ | $0_{\pm0}$ | $0_{\pm0}$ | $2_{\pm2}$ | $8_{\pm11}$ | $88_{\pm8}$ |
| | overall | $0_{\pm0}$ | $20_{\pm0}$ | $1_{\pm1}$ | $19_{\pm1}$ | $0_{\pm0}$ | $8_{\pm5}$ | $0_{\pm0}$ | $0_{\pm0}$ | $0_{\pm0}$ | $19_{\pm1}$ | $47_{\pm7}$ | $97_{\pm1}$ |
| puzzle-4x6 | task1 | $0_{\pm0}$ | $87_{\pm9}$ | $0_{\pm0}$ | $48_{\pm39}$ | $0_{\pm0}$ | $37_{\pm26}$ | $0_{\pm0}$ | $0_{\pm0}$ | $0_{\pm0}$ | $61_{\pm16}$ | $98_{\pm4}$ | $100_{\pm0}$ |
| | task2 | $0_{\pm0}$ | $0_{\pm0}$ | $0_{\pm0}$ | $5_{\pm10}$ | $0_{\pm0}$ | $3_{\pm7}$ | $0_{\pm0}$ | $0_{\pm0}$ | $0_{\pm0}$ | $2_{\pm2}$ | $46_{\pm30}$ | $66_{\pm13}$ |
| | task3 | $0_{\pm0}$ | $0_{\pm0}$ | $0_{\pm0}$ | $0_{\pm0}$ | $0_{\pm0}$ | $0_{\pm0}$ | $0_{\pm0}$ | $0_{\pm0}$ | $0_{\pm0}$ | $0_{\pm0}$ | $34_{\pm10}$ | $67_{\pm21}$ |
| | task4 | $0_{\pm0}$ | $0_{\pm0}$ | $0_{\pm0}$ | $0_{\pm0}$ | $0_{\pm0}$ | $0_{\pm0}$ | $0_{\pm0}$ | $0_{\pm0}$ | $0_{\pm0}$ | $1_{\pm1}$ | $5_{\pm6}$ | $23_{\pm7}$ |
| | task5 | $0_{\pm0}$ | $0_{\pm0}$ | $0_{\pm0}$ | $0_{\pm0}$ | $0_{\pm0}$ | $0_{\pm0}$ | $0_{\pm0}$ | $0_{\pm0}$ | $0_{\pm0}$ | $0_{\pm0}$ | $0_{\pm0}$ | $0_{\pm0}$ |
| | overall | $0_{\pm0}$ | $17_{\pm2}$ | $0_{\pm0}$ | $11_{\pm8}$ | $0_{\pm0}$ | $8_{\pm5}$ | $0_{\pm0}$ | $0_{\pm0}$ | $0_{\pm0}$ | $13_{\pm3}$ | $37_{\pm4}$ | $51_{\pm5}$ |

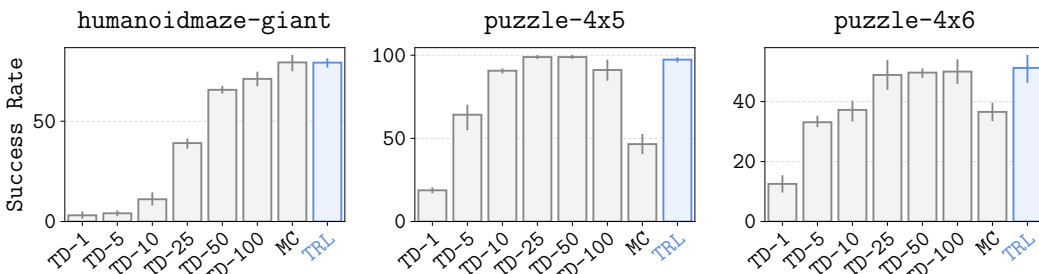

Figure 3: **TRL matches the best, individually tuned TD-$n$ baseline, without needing to set $n$.**

**Methods.** In these environments, we mainly compare the divide-and-conquer value update rule of TRL with existing value learning paradigms.

- **TRL** (ours), which employs divide-and-conquer value learning (Equation (12)).
- **TD** and **TD-$n$**, which employ ($n$-step) temporal difference value learning (Section C.1).
- **MC**, which employs Monte Carlo value learning (Equation (2)).

To conduct a controlled experiment that solely focuses on the value update rules, we fix the other hyperparameters, *e.g.*, goal hindsight relabeling ratios and policy extraction methods, except for the BC coefficient $\alpha$ in DDPG+BC (Equation (5)), which we individually tune for each method and task. This makes the results fully compatible, enabling an apples-to-apples comparison of different value learning paradigms.

Additionally, for reference, we also evaluate several standard offline GCRL algorithms on these long-horizon tasks. We consider four methods considered in the work by Park et al. (2025b), flow BC (FBC) (Chi et al., 2023), IQL (Kostrikov et al., 2022), CRL (Eysenbach et al., 2022), SAC+BC (Park et al., 2025b), WGCSL (Yang et al., 2022), and HIQL (Park et al., 2023), and three existing methods that employ the soft or hard triangle inequality, QRL (Wang et al., 2023), TDP (Kaelbling, 1993; Dhiman et al., 2018; Jurgenson et al., 2020), and COE (Piekos et al., 2023). Among them, IQL, SAC+BC, WGCSL, and HIQL are based on TD learning, and CRL is based on MC learning.

**Results.** We present the main comparison results in Table 1 and the comparison results with TD-$\{1, 5, 10, 25, 50, 100\}$ in Figure 3. The results suggest that TRL achieves the best performance across all three environments, outperforming or matching both TD- and MC-based approaches as well as other previous GCRL methods. Notably, TRL is the only triangle inequality-based method that achieves non-trivial performance on these challenging tasks.

In particular, we highlight that 1-step TD learning struggles on long-horizon tasks. While TD-$n$ or MC value learning can improve performance, Figure 3 indicates that it requires careful, task-

dependent tuning of $n$. In contrast, TRL achieves the best performance **without needing to set $n$**, matching the performance of the best TD-$n$ baseline individually tuned for each task. This highlights the benefits of our divide-and-conquer value learning framework.

## 5.2 How does TRL compare to prior methods on standard benchmarks?

Next, we compare the performance of TRL with existing offline GCRL algorithms on OG-Bench (Park et al., 2025a), a standard benchmark in offline goal-conditioned RL. The main research question here is whether TRL is comparable to existing algorithms on **regular, not necessarily long-horizon** tasks.

**Tasks.** We consider 10 goal-reaching tasks from OGBench. They span across robotic maze navigation (`{point, ant, humanoid}maze`), ball control (`antsoccer`), robotic object manipulation (`cube, scene`) and puzzle solving (`puzzle`). We use the `oraclerep` variants of these tasks, as in Park et al. (2025b). For datasets, we use the standard `navigate` and `play` datasets consisting of task-agnostic trajectories that randomly navigate the maze or perform random tasks.

**Methods.** We consider five widely used offline GCRL methods in this section: BC, FBC, IVL (Kostrikov et al., 2022; Park et al., 2023), IQL (Kostrikov et al., 2022), CRL (Eysenbach et al., 2022), and QRL (Wang et al., 2023). As in the previous section, we additionally consider two algorithms that employ the soft triangle inequality, TDP (Kaelbling, 1993; Dhiman et al., 2018; Jurgenson et al., 2020) and COE (Piekos et al., 2023).

OGBench Performance (10 Environments, 50 Tasks)

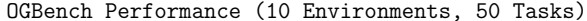

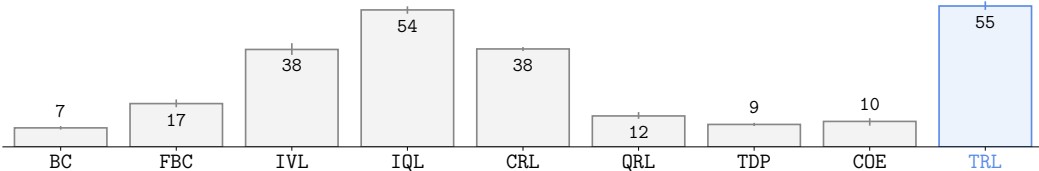

Figure 4: **TRL achieves strong performance on standard OGBench tasks.** While TRL is not specifically designed for short-horizon tasks, it outperforms or matches previous GCRL methods on a standard benchmark.

**Results.** Figure 4 shows the aggregated performance of each method over 10 OGBench environments and 50 evaluation tasks (see Table 2 for the full table). The results suggest that TRL achieves the best performance on average. While the gap between TRL and the second-best method (IQL) is narrower than that in Section 5.1, this is as expected to some extent, since TRL has its strongest advantages in long-horizon tasks. We also note that TRL is the only triangle inequality-based method that achieves strong performance on these robotic tasks.

## 5.3 Ablation Studies

We present ablation studies on three components of TRL (expectile $\kappa$, subgoal distributions, and distance-based re-weighting factor $\lambda$) in this section.

**Expectile $\kappa$.** TRL uses expectile regression to approximate the $\max$ operator in Equation (9). We evaluate TRL with $\kappa \in \{0.5, 0.7\}$, and present the results in Figure 5. The results show that while $\kappa = 0.5$ (*i.e.*, behavioral regression) works well in `humanoidmaze-medium` and `scene-play`, $\kappa > 0.5$ is crucial for achieving strong performance on `humanoidmaze-giant`.

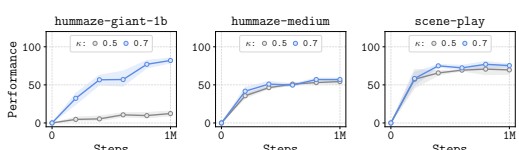

Figure 5: **Ablation study on the expectile $\kappa$.**

The strong performance with $\kappa = 0.5$ on some tasks is akin to how one-step RL (*i.e.*, behavioral value learning) is often enough to achieve solid performance in offline RL (Brandfonbrener et al., 2021; Park et al., 2025b).

**Subgoal distributions.** Another key feature of TRL is the use of *behavioral* (in-trajectory) subgoals. In Figure 6, we ablate this choice by comparing behavioral subgoals ("in-traj") with random subgoals ("random"). The results suggest that random subgoals substantially degrade performance, highlighting the importance of using behavioral subgoals.

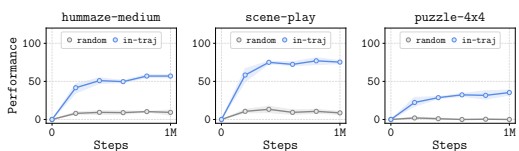

Figure 6: **Ablation study on subgoal distributions.**

**Distance-based re-weighting factor $\lambda$.** Figure 7 shows an ablation study on TRL's distance-based re-weighting factor $\lambda$. Note that a positive $\lambda$ makes TRL focus more on shorter trajectory chunks. The results show that enabling re-weighting (*i.e.*, $\lambda > 0$) often leads to better performance.

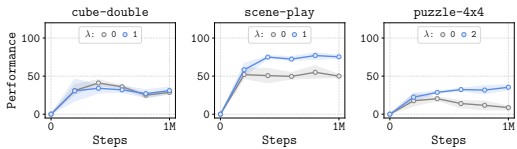

Figure 7: **Ablation study on the weighting factor $\lambda$.**

## 6 WHAT'S NEXT?

In this work, we have shown that Transitive RL exhibits better horizon scalability than standard TD- or MC-based value learning approaches in offline goal-conditioned RL. At a high level, this provides a piece of affirmative evidence for the hypothesis we posed in Section 1: namely, a *divide-and-conquer* paradigm may potentially lead to an ideal goal-conditioned value learning method that is free from the curse of horizon.

This is only the beginning of the journey. For example, it remains an open question whether a similar divide-and-conquer value learning technique could be applied to learn an *unbiased* value function in stochastic environments (which is a limitation of TRL, as well as many other works leveraging the vanilla triangle inequality in GCRL). The successor temporal distance framework proposed by Myers et al. (2024) may provide clues to this question. Another open question is whether TRL (or any divide-and-conquer-style algorithm) can be extended to general reward-based RL tasks, beyond goal-conditioned RL. We hope the ideas and techniques introduced in this work help address these important open questions and facilitate future progress toward scalable value learning algorithms.

## ACKNOWLEDGMENT

This work was partly supported by the Korea Foundation for Advanced Studies (KFAS), the NSF Graduate Research Fellowship, AFOSR FA9550-22-1-0273, and ONR N00014-22-1-2773. This research used the Savio computational cluster resource provided by the Berkeley Research Computing program at UC Berkeley.

## REPRODUCIBILITY STATEMENT

We provide the code at https://github.com/aoberai/trl and describe the full experimental details in Section C.

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

## A  PROOFS

In this section, we show that TRL's random sampling variant of divide-and-conquer value learning still gives us a logarithmic dependency on the horizon in the ideal, tabular case. Let $B(n)$ be the expected number of *maximum* Bellman recursions to compute the value $V(s_i, s_j)$ for a length-$n$ trajectory chunk, $(s_i, s_{i+1}, \ldots, s_{j=i+n})$ with the random sampling variant of transitive Bellman updates. Note that $B$ satisfies the following recursive equation:

$$B(1) = 0,$$

$$B(n) = 1 + \frac{1}{n-1} \sum_{k=1}^{n-1} \max(B(k), B(n-k)),$$

for $n > 1$.

Then, we have the following results:

**Lemma A.1.** *Let $n \geq 2$ be an integer and $C(n) = \frac{1}{n-1} \sum_{k=1}^{n-1} \max(k, n-k)$. Then $C(n) \leq 3n/4$.*

*Proof.* If $n = 2m$ (even),

$$\begin{aligned}
C(n) &= \frac{1}{n-1} \sum_{k=1}^{n-1} \max(k, n-k) \\
&= \frac{1}{2m-1}(m + 2((m+1) + \cdots + (2m-1))) \\
&= \frac{1}{2m-1}(m + 3m(m-1)) \\
&= \frac{3m^2 - 2m}{2m-1} \\
&\leq \frac{3}{2}m \\
&= \frac{3}{4}n.
\end{aligned}$$

If $n = 2m + 1$ (odd),

$$\begin{aligned}
C(n) &= \frac{1}{n-1} \sum_{k=1}^{n-1} \max(k, n-k) \\
&= \frac{1}{2m}(2((m+1) + \cdots + 2m)) \\
&= \frac{1}{2m}(m(3m+1)) \\
&= \frac{3m+1}{2} \\
&\leq \frac{6m+3}{4} \\
&= \frac{3}{4}n.
\end{aligned}$$

$\square$

**Theorem A.2.** $B(n) \leq \log n / \log(4/3)$.

*Proof.* Recall that we have the following recursive equation:

$$B(1) = 0,$$

$$B(n) = 1 + \frac{1}{n-1} \sum_{k=1}^{n-1} \max(B(k), B(n-k)),$$

for $n > 1$.

First, note that $B(1) = 0 \leq \log 1 / \log(4/3)$. For $n \geq 2$, we proceed by induction, assuming that $B(m) \leq \log m / \log(4/3)$ for all $m < n$. Then,

$$
\begin{aligned}
B(n) &= 1 + \frac{1}{n-1} \sum_{k=1}^{n-1} \max(B(k), B(n-k)) \\
&\leq 1 + \frac{1}{n-1} \sum_{k=1}^{n-1} \frac{\max(\log k, \log(n-k))}{\log(4/3)} \\
&= 1 + \frac{1}{\log(4/3)} \frac{1}{n-1} \sum_{k=1}^{n-1} \log(\max(k, n-k)) \\
&\leq 1 + \frac{1}{\log(4/3)} \log\left( \frac{1}{n-1} \sum_{k=1}^{n-1} \max(k, n-k) \right) \\
&= 1 + \frac{1}{\log(4/3)} \log C(n) \\
&\leq 1 + \frac{1}{\log(4/3)} \log\left( \frac{3}{4} n \right) \\
&= \frac{\log n}{\log(4/3)},
\end{aligned}
$$

where we use the inductive hypothesis in the first inequality, Jensen's inequality in the second, and Theorem A.1 in the third. $\qquad\square$

Theorem A.2 shows that the number of expected maximum Bellman recursions in the random sampling variant of transitive Bellman updates also has a logarithmic dependency on the horizon length.

# B ADDITIONAL RESULTS

We provide the full comparison results on the standard OGBench tasks (Section 5.2) in Table 2.

Table 2: **Full results on standard OGBench tasks.**

| Environment | Task | BC | FBC | IVL | IQL | CRL | QRL | TDP | COE | TRL |
|---|---|---|---|---|---|---|---|---|---|---|
| pointmaze-large-navigate-oraclerep-v0 | task1 | 53 ±24 | 98 ±3 | 94 ±6 | 92 ±9 | 23 ±18 | 31 ±28 | 40 ±19 | 67 ±26 | 52 ±19 |
| | task2 | 3 ±4 | 17 ±6 | 0 ±0 | 0 ±0 | 42 ±28 | 5 ±6 | 0 ±0 | 0 ±0 | 1 ±2 |
| | task3 | 17 ±9 | 76 ±11 | 100 ±0 | 76 ±13 | 67 ±14 | 0 ±0 | 4 ±4 | 12 ±8 | 7 ±5 |
| | task4 | 23 ±10 | 79 ±8 | 3 ±5 | 0 ±0 | 13 ±18 | 1 ±2 | 56 ±12 | 44 ±27 | 41 ±13 |
| | task5 | 30 ±22 | 87 ±5 | 44 ±39 | 0 ±0 | 21 ±8 | 0 ±0 | 49 ±16 | 48 ±16 | 64 ±14 |
| | overall | 25 ±3 | 71 ±4 | 48 ±10 | 34 ±3 | 33 ±7 | 7 ±7 | 30 ±5 | 34 ±7 | 33 ±5 |
| antmaze-large-navigate-oraclerep-v0 | task1 | 3 ±3 | 13 ±11 | 11 ±13 | 23 ±13 | 87 ±9 | 51 ±15 | 6 ±5 | 6 ±6 | 57 ±13 |
| | task2 | 22 ±12 | 24 ±8 | 19 ±5 | 44 ±20 | 64 ±25 | 69 ±12 | 23 ±6 | 26 ±11 | 66 ±4 |
| | task3 | 53 ±10 | 48 ±6 | 55 ±6 | 82 ±6 | 91 ±3 | 96 ±3 | 69 ±9 | 67 ±4 | 80 ±4 |
| | task4 | 17 ±9 | 7 ±3 | 4 ±3 | 15 ±7 | 92 ±4 | 57 ±16 | 17 ±10 | 14 ±1 | 8 ±8 |
| | task5 | 17 ±5 | 16 ±4 | 18 ±12 | 20 ±7 | 90 ±6 | 61 ±12 | 17 ±4 | 18 ±5 | 18 ±11 |
| | overall | 22 ±4 | 22 ±4 | 21 ±6 | 37 ±8 | 85 ±7 | 67 ±8 | 27 ±3 | 26 ±3 | 46 ±5 |
| humanoidmaze-medium-navigate-oraclerep-v0 | task1 | 10 ±7 | 7 ±9 | 29 ±4 | 31 ±9 | 91 ±8 | 8 ±12 | 4 ±3 | 6 ±4 | 76 ±6 |
| | task2 | 6 ±4 | 8 ±5 | 37 ±12 | 82 ±9 | 96 ±1 | 22 ±29 | 7 ±3 | 9 ±2 | 96 ±2 |
| | task3 | 8 ±5 | 15 ±5 | 16 ±7 | 6 ±6 | 72 ±17 | 30 ±21 | 12 ±3 | 13 ±3 | 8 ±8 |
| | task4 | 2 ±1 | 3 ±3 | 0 ±0 | 0 ±0 | 8 ±7 | 9 ±9 | 0 ±0 | 1 ±1 | 11 ±3 |
| | task5 | 11 ±5 | 12 ±2 | 39 ±16 | 63 ±12 | 93 ±4 | 20 ±19 | 14 ±4 | 14 ±7 | 92 ±3 |
| | overall | 7 ±2 | 9 ±3 | 24 ±4 | 36 ±3 | 72 ±5 | 18 ±17 | 7 ±0 | 9 ±2 | 57 ±1 |
| humanoidmaze-large-navigate-oraclerep-v0 | task1 | 1 ±1 | 0 ±0 | 9 ±3 | 10 ±1 | 49 ±17 | 3 ±3 | 0 ±0 | 0 ±0 | 9 ±6 |
| | task2 | 0 ±0 | 0 ±0 | 0 ±0 | 0 ±0 | 1 ±1 | 0 ±0 | 0 ±0 | 0 ±0 | 0 ±0 |
| | task3 | 3 ±1 | 2 ±1 | 3 ±2 | 11 ±6 | 63 ±25 | 9 ±6 | 3 ±4 | 4 ±3 | 29 ±9 |
| | task4 | 3 ±3 | 1 ±1 | 1 ±1 | 2 ±1 | 9 ±8 | 0 ±0 | 2 ±2 | 4 ±2 | 4 ±4 |
| | task5 | 2 ±2 | 2 ±1 | 2 ±1 | 0 ±0 | 20 ±17 | 1 ±1 | 0 ±0 | 2 ±2 | 0 ±0 |
| | overall | 2 ±1 | 1 ±1 | 3 ±1 | 5 ±1 | 28 ±5 | 3 ±2 | 1 ±1 | 2 ±0 | 8 ±1 |
| antsoccer-arena-navigate-oraclerep-v0 | task1 | 7 ±4 | 30 ±4 | 76 ±5 | 86 ±6 | 52 ±11 | 20 ±5 | 12 ±1 | 6 ±5 | 89 ±3 |
| | task2 | 6 ±3 | 24 ±7 | 62 ±5 | 92 ±4 | 36 ±4 | 21 ±3 | 7 ±6 | 11 ±2 | 85 ±5 |
| | task3 | 2 ±1 | 14 ±6 | 82 ±8 | 87 ±5 | 49 ±10 | 13 ±6 | 1 ±2 | 4 ±3 | 89 ±5 |
| | task4 | 2 ±1 | 11 ±7 | 34 ±6 | 59 ±2 | 13 ±7 | 3 ±3 | 3 ±1 | 2 ±1 | 48 ±12 |
| | task5 | 1 ±1 | 6 ±3 | 54 ±4 | 63 ±7 | 22 ±8 | 8 ±3 | 3 ±1 | 3 ±1 | 53 ±3 |
| | overall | 3 ±1 | 17 ±2 | 62 ±2 | 77 ±3 | 35 ±5 | 13 ±1 | 5 ±1 | 5 ±2 | 73 ±4 |
| cube-single-play-oraclerep-v0 | task1 | 7 ±9 | 17 ±5 | 87 ±3 | 97 ±3 | 64 ±6 | 6 ±7 | 4 ±3 | 7 ±7 | 98 ±2 |
| | task2 | 8 ±6 | 18 ±12 | 93 ±4 | 96 ±3 | 61 ±8 | 8 ±4 | 5 ±4 | 6 ±12 | 97 ±3 |
| | task3 | 9 ±6 | 22 ±5 | 93 ±2 | 99 ±1 | 69 ±9 | 9 ±3 | 1 ±2 | 2 ±3 | 99 ±1 |
| | task4 | 6 ±2 | 20 ±6 | 85 ±7 | 92 ±4 | 56 ±10 | 2 ±2 | 3 ±2 | 27 ±17 | 93 ±6 |
| | task5 | 5 ±5 | 14 ±7 | 82 ±4 | 89 ±5 | 66 ±18 | 3 ±2 | 0 ±0 | 24 ±21 | 87 ±7 |
| | overall | 7 ±2 | 18 ±5 | 88 ±2 | 95 ±1 | 63 ±5 | 6 ±3 | 3 ±1 | 13 ±8 | 95 ±2 |
| cube-double-play-oraclerep-v0 | task1 | 6 ±5 | 17 ±8 | 88 ±4 | 92 ±6 | 77 ±8 | 7 ±4 | 6 ±2 | 1 ±2 | 73 ±5 |
| | task2 | 0 ±0 | 1 ±1 | 78 ±5 | 84 ±7 | 42 ±9 | 0 ±0 | 0 ±0 | 0 ±0 | 23 ±7 |
| | task3 | 0 ±0 | 0 ±0 | 75 ±6 | 85 ±2 | 39 ±10 | 0 ±0 | 0 ±0 | 0 ±0 | 30 ±11 |
| | task4 | 0 ±0 | 1 ±2 | 8 ±5 | 12 ±9 | 1 ±1 | 0 ±0 | 0 ±0 | 0 ±0 | 3 ±3 |
| | task5 | 0 ±0 | 2 ±2 | 47 ±14 | 45 ±8 | 17 ±4 | 0 ±0 | 0 ±0 | 0 ±0 | 18 ±7 |
| | overall | 1 ±1 | 4 ±2 | 59 ±2 | 64 ±4 | 35 ±3 | 1 ±1 | 1 ±0 | 0 ±0 | 30 ±5 |
| scene-play-oraclerep-v0 | task1 | 14 ±8 | 60 ±6 | 97 ±3 | 99 ±1 | 71 ±13 | 19 ±8 | 43 ±4 | 32 ±8 | 97 ±2 |
| | task2 | 2 ±1 | 13 ±3 | 92 ±4 | 96 ±2 | 14 ±4 | 2 ±2 | 9 ±2 | 2 ±2 | 95 ±3 |
| | task3 | 2 ±3 | 21 ±9 | 83 ±8 | 89 ±8 | 33 ±9 | 1 ±1 | 3 ±4 | 2 ±3 | 97 ±3 |
| | task4 | 3 ±1 | 19 ±8 | 43 ±28 | 13 ±11 | 12 ±3 | 6 ±3 | 2 ±2 | 3 ±1 | 76 ±17 |
| | task5 | 0 ±0 | 3 ±4 | 23 ±16 | 10 ±9 | 2 ±2 | 1 ±1 | 0 ±0 | 0 ±0 | 18 ±9 |
| | overall | 4 ±2 | 23 ±2 | 68 ±10 | 61 ±3 | 26 ±5 | 6 ±2 | 12 ±1 | 8 ±2 | 77 ±2 |
| puzzle-3x3-play-oraclerep-v0 | task1 | 4 ±4 | 10 ±1 | 4 ±2 | 99 ±1 | 15 ±4 | 3 ±3 | 5 ±1 | 7 ±3 | 99 ±1 |
| | task2 | 1 ±2 | 1 ±1 | 2 ±2 | 99 ±1 | 6 ±3 | 0 ±0 | 1 ±1 | 2 ±2 | 99 ±1 |
| | task3 | 1 ±1 | 1 ±1 | 2 ±1 | 99 ±1 | 1 ±1 | 0 ±0 | 0 ±0 | 1 ±2 | 100 ±0 |
| | task4 | 0 ±0 | 1 ±1 | 1 ±1 | 98 ±1 | 1 ±1 | 0 ±0 | 2 ±3 | 1 ±1 | 98 ±1 |
| | task5 | 1 ±1 | 2 ±2 | 1 ±1 | 95 ±2 | 2 ±2 | 0 ±0 | 1 ±1 | 0 ±0 | 99 ±1 |
| | overall | 1 ±1 | 3 ±1 | 2 ±1 | 98 ±0 | 5 ±1 | 1 ±1 | 2 ±0 | 2 ±0 | 99 ±0 |
| puzzle-4x4-play-oraclerep-v0 | task1 | 1 ±1 | 1 ±1 | 6 ±4 | 33 ±14 | 1 ±1 | 0 ±0 | 1 ±1 | 0 ±0 | 47 ±5 |
| | task2 | 0 ±0 | 1 ±1 | 4 ±4 | 0 ±0 | 0 ±0 | 0 ±0 | 0 ±0 | 1 ±1 | 17 ±5 |
| | task3 | 0 ±0 | 1 ±1 | 6 ±1 | 58 ±10 | 1 ±1 | 0 ±0 | 1 ±1 | 0 ±0 | 38 ±13 |
| | task4 | 1 ±1 | 1 ±1 | 6 ±6 | 22 ±5 | 1 ±1 | 0 ±0 | 0 ±0 | 0 ±0 | 34 ±2 |
| | task5 | 0 ±0 | 0 ±0 | 6 ±1 | 29 ±9 | 0 ±0 | 0 ±0 | 1 ±1 | 0 ±0 | 32 ±6 |
| | overall | 0 ±0 | 1 ±0 | 5 ±2 | 28 ±4 | 0 ±0 | 0 ±0 | 0 ±0 | 0 ±0 | 34 ±4 |

## C  EXPERIMENTAL DETAILS

In this section, we describe the full details of our experiments. We provide the code and instructions at https://github.com/aoberai/trl.

### C.1  METHODS

Here, we describe the previous offline GCRL algorithms considered in this work.

**BC, FBC, IVL (Kostrikov et al., 2022; Park et al., 2023), IQL (Kostrikov et al., 2022), CRL (Eysenbach et al., 2022), HIQL (Park et al., 2023), and QRL (Wang et al., 2023).** We employ the original implementations by Park et al. (2025a;b), and we refer to these works for full details.

**WGCSL (Yang et al., 2022).** We use our JAX (Bradbury et al., 2018) re-implementation of the original implementation of WGCSL. We gradually change the advantage percentile from 0 to 80, and use a discounted relabeling weight of 0.999 and a default advantage weight $\epsilon_{\min}$ of 0.05.

**TD and TD-$n$.** The TD (which is equivalent to TD-1) and TD-$n$ baselines used in Section 5.1 minimize the following IQL-like objective:

$$L^0(Q) = \mathbb{E}_{\tau \sim \mathcal{D}} \left[ D \left( Q(s_i, a_i, s_i), \gamma^0 \right) \right], \tag{13}$$

$$L^1(Q) = \mathbb{E}_{\tau \sim \mathcal{D}} \left[ D_\kappa \left( Q(s_i, a_i, s_j), \gamma^n \bar{Q}(s_{i+n}, a_{i+n}, s_j) \right) \right], \tag{14}$$

$$L^{\mathrm{TD}-n}(Q) = L^0(Q) + L^1(Q), \tag{15}$$

where trajectories $\tau$ are sampled from the same trajectory distribution used in TRL. When $i + n > j$, we appropriately clip the value of $n$ (this is omitted in the objective above for notational simplicity). Intuitively, the objective above can be viewed as the closest TD-based variant (ablation) of TRL.

**TDP (Kaelbling, 1993; Dhiman et al., 2018; Jurgenson et al., 2020).** We refer to triangle-inequality dynamic programming (TDP) as GCRL algorithms that take a hard maximum over the entire state space to compute an optimal subgoal (Kaelbling, 1993; Dhiman et al., 2018; Jurgenson et al., 2020). These algorithms were originally designed for tabular environments, so we consider a sampling-based variant of TDP for our experiments. Specifically, to compute a hard maximum in our continuous-control environments, we sample $M$ subgoal candidates from the dataset, and take the maximum over these $M$ states. Formally, we minimize the following loss:

$$L^0(Q) = \mathbb{E}_{s,a \sim \mathcal{D}} \left[ D \left( Q(s, a, s), \gamma^0 \right) \right], \tag{16}$$

$$L^1(Q) = \mathbb{E}_{s,a,s' \sim \mathcal{D}} \left[ D \left( Q(s, a, s'), \gamma^1 \right) \right], \tag{17}$$

$$L^\infty(Q) = \mathbb{E}_{s,a,g^r \sim \mathcal{D}} \left[ D \left( Q(s, a, g^r), \gamma^P \right) \right], \tag{18}$$

$$L^\triangle(Q) = \mathbb{E}_{s,a,g,W \sim \mathcal{D}} \left[ D \left( Q(s, a, g), \max_{(w, a_w) \in W} \bar{Q}(s, a, w) \bar{Q}(w, a_w, g) \right) \right], \tag{19}$$

$$L^{\mathrm{TDP}}(Q) = L^0(Q) + L^1(Q) + L^\infty(Q) + L^\triangle(Q), \tag{20}$$

where $g^r$ is a randomly sampled goal from the dataset (note that this distribution is different from that of $g$, as $g$ is typically partially sampled with hindsight relabeling), $P$ is a tunable hyperparameter (a large number to approximate the distance between a random $(s, g)$ pair (Jurgenson et al., 2020)), and $W = \{(w^{(i)}, a_w^{(i)})\}_{i=1}^M$ is a set of $M$ randomly sampled state-action pairs from the dataset.

**COE (Piekos et al., 2023).** COE is a triangle inequality-based GCRL algorithm that uses a separate generator network $G(s, a, g) : \mathcal{S} \times \mathcal{A} \times \mathcal{S} \to \mathcal{S}$ to predict the optimal subgoal. Since this algorithm was originally designed for online GCRL, we add an additional behavioral regularizer to constrain this generator network to produce in-distribution states, as commonly done in offline RL (Wu et al., 2019; Fujimoto & Gu, 2021; Tarasov et al., 2023). Formally, we minimize the following losses to train the value function and generator:

$$L^1(Q) = \mathbb{E}_{s,a,s' \sim \mathcal{D}} \left[ D \left( Q(s, a, s'), \gamma^1 \right) \right], \tag{21}$$

$$L^\triangle(Q) = \mathbb{E}_{\substack{s,a,g \sim \mathcal{D}, \\ w = G(s,a,g)}} \left[ D \left( Q(s, a, g), \bar{Q}(s, a, w) \bar{Q}(w, \pi(w, g), g) \right) \right], \tag{22}$$

$$L^{\mathrm{COE}}(Q) = L^1(Q) + L^\triangle(Q), \tag{23}$$

$$L^{\mathrm{COE}}(G) = \mathbb{E}_{\substack{s,a,g,g^r \sim \mathcal{D}, \\ w = G(s,a,g)}} \left[ -\bar{Q}(s, a, w) \bar{Q}(w, \pi(w, g), g) - \beta \| w - g^r \|_2^2 \right], \tag{24}$$

where $g^r$ is a randomly sampled goal from the dataset, $\beta$ is a hyperparameter that controls the strength of the regularizer. In the above, we slightly abuse the notation by assuming that the goal-conditioned policy $\pi$ is deterministic.

## C.2 ORACLE DISTILLATION

In our experiments, we mainly employ the `oraclerep` variants, following Park et al. (2025b). In `oraclerep` environments, we are given an "oracle" representation of goals ($\varphi(g) : \mathcal{S} \to \mathcal{Z}$, where $\mathcal{Z}$ is an oracle goal representation space), which typically corresponds to a subset of the state dimensions. For example, in `humanoidmaze`, a full (goal) state is a 69-dimensional vector, but the corresponding oracle representation is a 2-dimensional vector consisting only of the $x$-$y$ coordinates. In these environments, we condition the policy and value functions on the oracle representation, not on the full goal state.

While we can easily incorporate this change in standard MC- and TD-based algorithms by simply parameterizing the policy and value functions with the oracle representation (*e.g.*, using $\pi(a \mid s, \varphi(g))$ instead of $\pi(a \mid s, g)$), it is not straightforward to use $\varphi$ in triangle inequality-based methods, such as QRL and TRL, because they assume that states and goals lie in the same space. To make such methods compatible with oracle representations, we apply a technique we call "oracle distillation." That is, we train an additional oracle representation-conditioned Q function, $Q^\varphi : \mathcal{S} \times \mathcal{A} \times \mathcal{Z} \to \mathbb{R}$, by distilling it from the original Q function, $Q : \mathcal{S} \times \mathcal{A} \times \mathcal{S} \to \mathbb{R}$, using the following loss:

$$L^{\text{distill}}(Q^\varphi) = \mathbb{E}_{s,a,g\sim\mathcal{D}}\left[D\left(Q^\varphi(s, a, \varphi(g)), Q(s, a, g)\right)\right]. \tag{25}$$

Here, the original Q function assumes that the state and goal spaces are the same and can therefore leverage the triangle inequality. After training $Q^\varphi$, we extract an oracle representation-conditioned policy from this distilled Q function.

## C.3 IMPLEMENTATION DETAILS

**Training and evaluation.** In our experiments, we train all agents for 1M gradient steps. For evaluation, we use 15 episodes for each of the five evaluation goals provided by OGBench (Park et al., 2025a). In tables, we report performance averaged over the last three evaluation epochs (*i.e.*, 800K, 900K, and 1M steps), following the protocol of OGBench. However, for the baselines from the work by Park et al. (2025b) (*i.e.*, FBC, IQL, CRL, SAC+BC, and HIQL in Table 1), we use the evaluation result only at the 1M epoch, as they were evaluated every 500K steps.

In Table 1, we take the performance of FBC, IQL, CRL, SAC+BC, and HIQL from the work by Park et al. (2025b). In Table 2, we re-run all baselines (BC, FBC, IVL, IQL, CRL, and QRL) using the original OGBench hyperparameters (Park et al., 2025a) on the `oraclerep` tasks (except for the value goal hindsight relabeling ratios, which we describe below). In these tables, we ensure apples-to-apples comparisons between TRL and other baselines by using the same default configurations (*e.g.*, we use the same network size, discount factor, training steps, etc.).

**Hyperparameters.** We provide the full list of hyperparameters in Tables 3 to 5. For the experiments in Section 5.1, we mostly follow the hyperparameters by Park et al. (2025b), and for the experiments in Section 5.2, we mostly follow the ones by Park et al. (2025a). In the hyperparameter tables, the tuple $(p^{\mathcal{D}}_{\text{cur}}, p^{\mathcal{D}}_{\text{geom}}, p^{\mathcal{D}}_{\text{traj}}, p^{\mathcal{D}}_{\text{rand}})$ denotes the hindsight goal relabeling ratios described in OGBench (Park et al., 2025a).

Table 3: **Hyperparameters for long-horizon OGBench tasks (Table 1).**

| Hyperparameter | Value |
|---|---|
| Gradient steps | $10^6$ |
| Optimizer | Adam (Kingma & Ba, 2015) |
| Learning rate | 0.0003 |
| Batch size | 1024 |
| MLP size | $[1024, 1024, 1024, 1024]$ |
| Nonlinearity | GELU (Hendrycks & Gimpel, 2016) |
| Target network update rate | 0.005 |
| Discount factor $\gamma$ | 0.999 |
| Policy $(p_{\text{cur}}^{\mathcal{D}}, p_{\text{geom}}^{\mathcal{D}}, p_{\text{traj}}^{\mathcal{D}}, p_{\text{rand}}^{\mathcal{D}})$ ratio | $(0,0,1,0)$ (humanoidmaze), $(0,0.5,0,0.5)$ (puzzle) |
| Value $(p_{\text{cur}}^{\mathcal{D}}, p_{\text{geom}}^{\mathcal{D}}, p_{\text{traj}}^{\mathcal{D}}, p_{\text{rand}}^{\mathcal{D}})$ ratio (TRL, TD, MC) | $(0,0,1,0)$ |
| Value $(p_{\text{cur}}^{\mathcal{D}}, p_{\text{geom}}^{\mathcal{D}}, p_{\text{traj}}^{\mathcal{D}}, p_{\text{rand}}^{\mathcal{D}})$ ratio (CRL) | $(0,1,0,0)$ |
| Value $(p_{\text{cur}}^{\mathcal{D}}, p_{\text{geom}}^{\mathcal{D}}, p_{\text{traj}}^{\mathcal{D}}, p_{\text{rand}}^{\mathcal{D}})$ ratio (others) | $(0.2, 0.5, 0, 0.3)$ |
| Policy extraction | Reparameterized gradients or rejection sampling (see Table 5) |

Table 4: **Hyperparameters for standard OGBench tasks (Table 2).**

| Hyperparameter | Value |
|---|---|
| Gradient steps | $10^6$ |
| Optimizer | Adam (Kingma & Ba, 2015) |
| Learning rate | 0.0003 |
| Batch size | 1024 |
| MLP size | $[512, 512, 512]$ |
| Nonlinearity | GELU (Hendrycks & Gimpel, 2016) |
| Target network update rate | 0.005 |
| Discount factor $\gamma$ | 0.99 (default), 0.995 (humanoidmaze) |
| Policy $(p_{\text{cur}}^{\mathcal{D}}, p_{\text{geom}}^{\mathcal{D}}, p_{\text{traj}}^{\mathcal{D}}, p_{\text{rand}}^{\mathcal{D}})$ ratio | $(0,0,1,0)$ |
| Value $(p_{\text{cur}}^{\mathcal{D}}, p_{\text{geom}}^{\mathcal{D}}, p_{\text{traj}}^{\mathcal{D}}, p_{\text{rand}}^{\mathcal{D}})$ ratio (TRL, CRL) | $(0,1,0,0)$ |
| Value $(p_{\text{cur}}^{\mathcal{D}}, p_{\text{geom}}^{\mathcal{D}}, p_{\text{traj}}^{\mathcal{D}}, p_{\text{rand}}^{\mathcal{D}})$ ratio (others) | $(0.2, 0.5, 0, 0.3)$ |
| Policy extraction | Reparameterized gradients (see Table 5) |

Table 5: **Task-specific hyperparameters.** We describe task-specific hyperparameters below ($\alpha$: BC coefficient for reparameterized gradients, $N$: sample count for rejection sampling, $M$: subgoal count, $P$: random goal distance, $\beta$: goal regularization coefficient, $\kappa$: expectile, $\lambda$: distance-based re-weighting factor).

| Environment | QRL $\alpha$ | TDP $(\alpha, N)$ | TDP $M$ | TDP $P$ | COE $\alpha$ | COE $\beta$ | TD $(\alpha, N)$ | MC $(\alpha, N)$ | TRL $(\alpha, N)$ | TRL $\kappa$ | TRL $\lambda$ |
|---|---|---|---|---|---|---|---|---|---|---|---|
| humanoidmaze-giant | 0.001 | $(3,-)$ | 8 | 1000 | 1 | 0.5 | $(0.3,-)$ | $(0.3,-)$ | $(0.1,-)$ | 0.7 | 0 |
| puzzle-4x5 | 3 | $(-,32)$ | 8 | 1000 | 3 | 10 | $(-,32)$ | $(-,32)$ | $(-,32)$ | 0.7 | 0 |
| puzzle-4x6 | 3 | $(-,32)$ | 8 | 1000 | 3 | 10 | $(-,32)$ | $(-,32)$ | $(-,32)$ | 0.7 | 0 |
| pointmaze-large | 0.0003 | $(5,-)$ | 8 | 500 | 1 | 1 | | | $(10,-)$ | 0.7 | 0.7 |
| antmaze-large | 0.003 | $(10,-)$ | 8 | 100 | 3 | 10 | | | $(0.7,-)$ | 0.7 | 0 |
| humanoidmaze-medium | 0.001 | $(5,-)$ | 8 | 500 | 0.1 | 3 | | | $(0.1,-)$ | 0.7 | 0 |
| humanoidmaze-large | 0.001 | $(5,-)$ | 8 | 500 | 0.1 | 1 | | | $(0.1,-)$ | 0.7 | 0.1 |
| antsoccer-arena | 0.003 | $(10,-)$ | 8 | 200 | 0.3 | 1 | | | $(0.3,-)$ | 0.7 | 0.5 |
| cube-single | 0.3 | $(5,-)$ | 8 | 500 | 0.3 | 10 | | | $(1,-)$ | 0.7 | 0.7 |
| cube-double | 0.3 | $(5,-)$ | 8 | 500 | 0.3 | 10 | | | $(10,-)$ | 0.7 | 1 |
| scene | 0.3 | $(1,-)$ | 16 | 200 | 1 | 10 | | | $(1,-)$ | 0.7 | 1 |
| puzzle-3x3 | 0.3 | $(5,-)$ | 8 | 500 | 3 | 10 | | | $(2,-)$ | 0.7 | 0.5 |
| puzzle-4x4 | 0.3 | $(5,-)$ | 8 | 500 | 1 | 10 | | | $(2,-)$ | 0.7 | 2 |

