# OpenReview forum: "Transitive RL: Value Learning via Divide and Conquer"
_ICLR.cc/2026/Conference — ICLR 2026 Poster_

### Official Review · Reviewer_6EwX · 2025-10-17

**Soundness:** 2
**Presentation:** 1
**Contribution:** 2
**Rating:** 4
**Confidence:** 4

**Summary:**

This paper proposes a method called Transitive RL for offline goal-conditioned RL. It introduces a novel divide-and-conquer approach to learn a value function with reduced accumulated error. On the recent OGBench benchmark, it demonstrates superior performance compared to well-established baselines such as IQL and CRL.

**Strengths:**

1. This paper provides an insightful connection between $V(s, g)$ and $\max_w V(s, w) V(w, g)$. Accordingly, the authors derive a novel divide-and-conquer value function to replace the target in Eq. 10.

2. The proposed algorithm is simple and achieves strong performance on recently proposed benchmarks.

**Weaknesses:**

The authors’ main hypothesis is that a divide-and-conquer algorithm *might* provide a path to overcoming the curse of horizon (line 55). I would argue that this is not a well-defined hypothesis. It would be more appropriate to state that a divide-and-conquer value learning algorithm can mitigate the curse of horizon in offline goal-conditioned tasks. Accordingly, **the paper’s framing should be revised to emphasise the offline goal-conditioned setting, rather than starting broadly from off-policy RL**.

I find this submission interesting; however, the presentation and narrative could be improved. The paper lacks a smooth transition from the problem statement (and background) to the proposed idea, and then to the corresponding solution.

**Questions:**

1. In Table 1, TRL significantly outperforms the selected baselines. However, I encourage the authors to include more recent offline GCRL methods, such as HIQL [1] and WGCSL [2].

2. It would be preferable to combine Sections 5.1 and 5.2, as they largely overlap, and move the ablation studies to the main pages.

3. Section 5.1 shows that TRL performs well on large-scale, long-horizon tasks. However, this evidence alone is insufficient to demonstrate that the performance gain arises from the divide-and-conquer value learning. The authors should design more detailed experiments to verify that the value function learned by the proposed method is indeed more accurate and less prone to overestimation, as stated in line 256.

**References**

[1] Park, S., Ghosh, D., Eysenbach, B., \& Levine, S. (2023). HIQL: Offline Goal-Conditioned RL with Latent States as Actions. NeurIPS.

[2] Yang, R., Lu, Y., Li, W., Sun, H., Fang, M., Du, Y., … Zhang, C. (2022). Rethinking Goal-Conditioned Supervised Learning and Its Connection to Offline RL. ICLR.

---

> ### Author Response · Authors · 2025-11-20
>
> Thank you for the detailed review and constructive feedback about this work. We especially appreciate your comment about the paper's framing, which we have updated accordingly in the revised draft. We have also evaluated two more baselines (HIQL and WGCSL) and performed an additional value overestimation analysis. We believe that these updates address the major concerns raised in your review, but if there are any remaining questions, feel free to let us know.
>
> * **"The paper’s framing should be revised to emphasise the offline goal-conditioned setting, rather than starting broadly from off-policy RL."**
>
> Thanks for the suggestion, which we believe is a valid point. Following the suggestion, **we have extensively rewritten Section 1 to clearly state our research hypothesis** ("Our central hypothesis is that, in this offline goal-conditioned setting, a divide-and-conquer value learning algorithm can mitigate the curse of horizon more effectively than standard TD- or MC-based methods"). We have denoted the changes in red in the revised draft. Please feel free to let us know if you have any additional concerns about the paper's framing and motivation.
>
> * **Additional comparison with HIQL and WGCSL**
>
> Thank you for the question! First, we would like to note that this work mainly focuses on *value learning*, so we previously compared TRL mainly with methods that do not involve hierarchical policies (Footnote 4), as TRL can also be orthogonally combined with hierarchical policy extraction methods like HIQL. However, following your suggestion, we have additionally compared TRL with HIQL and WGCSL on the long-horizon tasks used in Section 5.1. Note that HIQL uses a hierarchical policy but uses a TD-1 value function.
>
> | Environment                   | SAC+BC     | QRL       | SGT       | COE       | TD         | MC                  | HIQL        | WGCSL     | TRL (ours)                 |
> |:------------------------------|:-----------|:----------|:----------|:----------|:-----------|:--------------------|:------------|:----------|:--------------------|
> | $\texttt{humanoidmaze-giant}$ | $5 \pm 0$  | $3 \pm 2$ | $2 \pm 0$ | $2 \pm 1$ | $3 \pm 2$  | $\mathbf{79} \pm 4$ | $23 \pm 13$ | $1 \pm 1$ | $\mathbf{79} \pm 2$ |
> | $\texttt{puzzle-4x5}$         | $19 \pm 1$ | $0 \pm 0$ | $0 \pm 0$ | $0 \pm 0$ | $19 \pm 1$ | $47 \pm 7$          | $8 \pm 5$   | $0 \pm 0$ | $\mathbf{97} \pm 1$ |
> | $\texttt{puzzle-4x6}$         | $11 \pm 8$ | $0 \pm 0$ | $0 \pm 0$ | $0 \pm 0$ | $13 \pm 3$ | $37 \pm 4$          | $8 \pm 5$   | $0 \pm 0$ | $\mathbf{51} \pm 5$ |
>
> The table above compares the performance of TRL and other (selected strong) baselines, including HIQL and WGCRL. The results suggest that both HIQL and WGCSL fall significantly behind TRL (we were unable to achieve non-trivial performance with WGCSL on the long-horizon puzzle tasks despite extensive tuning). We believe this is mainly because both HIQL and WGCSL are based on vanilla TD-1 value learning (even though HIQL uses a hierarchical policy), which suffers from bias accumulation in long-horizon environments, as discussed in the paper. In contrast, TRL's value learning is based on divide-and-conquer (rather than TD), which often scales better to long-horizon tasks. We discuss this point further in the next question.

---

> > ### Author Response · Authors · 2025-11-20
> >
> > * **How can we know the performance gain of TRL comes from divide-and-conquer value learning?**
> >
> > Thank you for raising this point. First, we would like to note that the TD, MC, and TRL methods in Section 5.1 **differ only in their value losses**. In particular, all these methods share the same network architecture, policy extraction method, and hindsight goal sampling strategy. Hence, the performance gain of TRL over these methods comes entirely from TRL's divide-and-conquer *value learning* paradigm.
> >
> > Moreover, following the suggestion, we further demonstrate how TRL's technique (in particular, in-sample maximization) mitigates value overestimation and leads to better performance. To do this, we evaluated an ablation of TRL where we replace in-sample maximization (expectile) with hard argmax (akin to prior work [1]) over $10$ randomly sampled in-trajectory subgoals.
> >
> > **[Average estimated Q values]**
> >
> > | Environment                    | TRL (ours)             | TRL (hard argmax)   |
> > |:-------------------------------|:----------------|:--------------------|
> > | $\texttt{pointmaze-large}$     | $0.55 \pm 0.00$ | $0.69 \pm 0.00$     |
> > | $\texttt{antmaze-large}$       | $0.49 \pm 0.00$ | $0.81 \pm 0.00$     |
> > | $\texttt{humanoidmaze-medium}$ | $0.64 \pm 0.01$ | $0.86 \pm 0.00$     |
> > | $\texttt{humanoidmaze-large}$  | $0.65 \pm 0.00$ | $0.85 \pm 0.00$     |
> > | $\texttt{antsoccer-arena}$     | $0.57 \pm 0.01$ | $0.83 \pm 0.00$     |
> > | $\texttt{cube-single}$         | $0.59 \pm 0.00$ | $0.79 \pm 0.01$     |
> > | $\texttt{cube-double}$         | $0.58 \pm 0.00$ | $0.77 \pm 0.01$     |
> > | $\texttt{scene}$               | $0.60 \pm 0.00$ | $0.78 \pm 0.00$     |
> > | $\texttt{puzzle-3x3}$          | $0.60 \pm 0.00$ | $0.77 \pm 0.00$     |
> > | $\texttt{puzzle-4x4}$          | $0.59 \pm 0.00$ | $0.98 \pm 0.00$     |
> >
> > **[Performance]**
> >
> > | Environment                    | TRL (ours)                 | TRL (hard argmax)   |
> > |:-------------------------------|:--------------------|:--------------------|
> > | $\texttt{pointmaze-large}$     | $\mathbf{33} \pm 5$ | $\mathbf{34} \pm 7$ |
> > | $\texttt{antmaze-large}$       | $\mathbf{46} \pm 5$ | $35 \pm 3$          |
> > | $\texttt{humanoidmaze-medium}$ | $\mathbf{57} \pm 1$ | $38 \pm 5$          |
> > | $\texttt{humanoidmaze-large}$  | $\mathbf{8} \pm 1$  | $6 \pm 3$           |
> > | $\texttt{antsoccer-arena}$     | $\mathbf{73} \pm 4$ | $41 \pm 1$          |
> > | $\texttt{cube-single}$         | $\mathbf{95} \pm 2$ | $88 \pm 7$          |
> > | $\texttt{cube-double}$         | $\mathbf{30} \pm 5$ | $5 \pm 2$           |
> > | $\texttt{scene}$               | $77 \pm 2$          | $\mathbf{83} \pm 8$ |
> > | $\texttt{puzzle-3x3}$          | $\mathbf{99} \pm 0$ | $63 \pm 3$          |
> > | $\texttt{puzzle-4x4}$          | $\mathbf{34} \pm 4$ | $0 \pm 1$           |
> >
> > The first table compares the average estimated Q values of TRL and its hard argmax variant, and the second compares their performances (averaged over four seeds). The results suggest that our in-sample maximization indeed effectively mitigates Q-value overestimation compared to hard argmax (as shown in the first table), and that this in turn leads to better performance (as shown in the second table).
> >
> > * **Restructuring the sections**
> >
> > Thanks for the suggestion! As we can use one more page for the rebuttal and camera-ready version, we have moved the ablation study to the main paper (Section 5.3).
> >
> > ---
> >
> > We would like to thank you again for raising important comments and questions about our work. We believe the revised framing and the additional experiments have strengthened the paper. Please let us know if you have any additional concerns or questions. If we have addressed your concerns, we would appreciate it if you could adjust the score accordingly.
> >
> > ---
> >
> > [1] Jurgenson et al., Sub-Goal Trees – a Framework for Goal-Based Reinforcement Learning (2020)

---

> ### Comment · Reviewer_6EwX · 2025-11-21
>
> Thank you for your reply. Most of my concerns have been addressed. I have the following suggestions:
>
> 1. Although Section 1 has been greatly improved, it could still be written in a more coherent and detailed way. For example:
>
>     a. The current explanation of offline GCRL may not be clear for new readers.
>
>     b. In line 58, you mention that the triangle inequality provides an additional structure that can be exploited. However, while Eq. 1 shows the inequality, its consequences are not explained. The equation alone does not illustrate the implications.
>
>     c. In the next paragraph, the authors could further explain their motivation for using a divide-and-conquer approach, and how this connects to the second paragraph.
>
>     d. In the final paragraph, please consider summarising the overall contributions more clearly and listing them explicitly.
>
> I acknowledge the contributions of the paper, and I genuinely hope the introduction can attract readers and effectively highlight these contributions. Additionally, if I were the authors, I would start from offline GCRL, talk about the curse of horizon in offline GCRL, discuss related work (one step, MC), propose divide-and-conquer idea, explain the method, and highlight the contributions. These are just my suggestions, not necessarily the only correct approach.
>
> 2. You may want to use a wrap-figure environment for the ablation studies.
>
> 3. Regarding the added baselines: while I understand that hierarchical policies are not the main focus of the paper, they are important reference points that help readers understand performance across all offline GCRL methods. I believe they are essential baselines, so please include the new baselines in Table 1.

---

> > ### Author Response · Authors · 2025-11-22
> >
> > We appreciate your prompt, detailed response as well as additional helpful suggestions. Following the suggestions, we have further revised Section 1 and added the HIQL and WGCSL results to Table 1. Please find our response below.
> >
> > * **Section 1**
> >
> > Thanks a lot for the detailed suggestions! We generally agree with them and **have further revised Section 1 accordingly**. Regarding the opening paragraphs, we would like to maintain the current structure because we believe (1) the challenge itself (the curse of horizon in off-policy RL) is not necessarily limited to offline goal-conditioned RL ([1] and Appendix D of [2]), and (2) we would like to potentially inspire a broader audience from the general off-policy RL community. This is especially because we believe the divide-and-conquer paradigm itself has the future potential to provide a scalable solution to general off-policy RL (we refer to our response to Reviewer EYPx for details regarding this point). Nonetheless, as we have clearly stated the scope and contributions of this work in both Section 1 and Abstract of the current draft, we expect that these opening paragraphs (simply) serve as high-level motivation and will not cause misunderstandings about the contributions of this work.
> >
> > * **`wrapfigure` for ablation studies**
> >
> > Thanks for the helpful suggestion. We have updated Section 5.3 accordingly.
> >
> > * **Additional baselines**
> >
> > Following the suggestion, we have added the new HIQL and WGCSL results to Table 1 and updated the Appendix.
> >
> > ---
> >
> > We would like to thank the reviewer again especially for the highly detailed feedback about Section 1 as well as the other helpful suggestions. Please feel free to let us know if you have any remaining concerns or questions.
> >
> > ---
> >
> > [1] Liu et al., Breaking the curse of horizon: Infinite-horizon off-policy estimation (2018) \
> > [2] Park et al., Horizon Reduction Makes RL Scalable (2025)

---

### Official Review · Reviewer_KdC6 · 2025-10-29

**Soundness:** 2
**Presentation:** 2
**Contribution:** 2
**Rating:** 4
**Confidence:** 3

**Summary:**

This paper introduces a method called TRL for GCRL. The idea is to long horizon tasks by formulating them as GCRL which then allows "dividing and conquering" to stitch together shorter plan segments to avoid the small errors that normally stack up and ruin long-term plans.

**Strengths:**

The idea seems sensible and the results seem strong. It seems to make the idea of triangle inequality for value learning work, although I am not very familiar with current GCRL literature.

**Weaknesses:**

**W1.** Complicated algorithm with lots of moving parts: eta-quantile, M subgoals, expectable loss, reweighting, separate "oracle distillation" network, policy extraction. This is many more "moving parts" than standard algorithms like IQL or TD-n, which could make it difficult to tune and implement.

**W2.** As far as I understand it relies on oracle representations (see Appendix D.2). This seems like a significant weakness.

**W3.** Structure: The triangle inequality is referred to in the abstract, introduction, related work (where there’s a whole subsection on it) etc, but only properly introduced in the middle of page 4. The triangle inequality really needs to be described earlier - in or before the dedicated discussion paragraph in related work.

**W4.** The paper tests on one suite of benchmarks. It would benefit from results on another set of benchmarks.

**W5.** Limited to GCRL and deterministic environments.

Other weaknesses: See questions.

**Questions:**

**Q1.** The authors state they take some baseline results from a prior paper (Table 1) but re-run others (Table 2). For the re-run baselines, how was fair tuning ensured? For example, the SGT-DP and COE baselines (Table 1) are also triangle-inequality methods that perform poorly - how can the authors be sure this is not due to a poor implementation or unfair tuning on their part?

**Q2.** The oracle distillation seems important but it is only mentioned in Appendix D.2. Could you do an ablation of this?

**Q3.** Using the BCE loss over the MSE loss seems unusual. Why was this chosen? Did you try MSE as well?

**Q4.** The authors claim "we experimentally find that behavioural subgoals can still be highly effective even when the dataset consists of uniformly random atomic motions". Which experiments does this refer to? It would strengthen the paper to have results across different dataset types - "random," "medium," and "expert" - to support this claim.

---

> ### Author Response · Authors · 2025-11-20
>
> Thank you for the detailed review and constructive feedback on this work. Following the suggestion, we have additionally evaluated several variants of TRL without oracle distillation and with the MSE loss, and clarified several points. We believe that these updates address the major concerns raised in your review, but if there are any remaining questions, feel free to let us know.
>
> * **Oracle distillation**
>
> We would like to first note that TRL does not *require* oracle distillation -- we used oracle distillation mainly because we followed the experimental setup in a previous work [1], which uses oracle representations. Moreover, *all* baselines considered in this work use the same oracle representation of goals (in both Sections 5.1 and 5.2), which ensures a fair comparison.
>
> To empirically show that TRL works well even without oracle distillation, we conducted an additional experiment *without* oracle representation/distillation on the three challenging, long-horizon tasks with 1B datasets (the ones used in Table 1).
>
> | Environment                   | TD (oraclerep)   | MC (oraclerep)   | TRL (oraclerep)   | TRL (w/o oraclerep)   |
> |:------------------------------|:-----------------|:-----------------|:------------------|:----------------------|
> | $\texttt{humanoidmaze-giant}$ | $3 \pm 2$        | $79 \pm 4$       | $79 \pm 2$        | $80 \pm 8$            |
> | $\texttt{puzzle-4x5}$         | $19 \pm 1$       | $47 \pm 7$       | $97 \pm 1$        | $80 \pm 8$            |
> | $\texttt{puzzle-4x6}$         | $13 \pm 3$       | $37 \pm 4$       | $51 \pm 5$        | $46 \pm 6$            |
>
> The table above compares TRL with and without oracle representations with the two closest baselines (TD and MC). The results suggest that, while using oracle representations/distillation (slightly) improves performance to some degree, **TRL still achieves strong performance without oracle representations**, even outperforming or at least matching TD and MC *with* oracle representations on these tasks.
>
> * **"TRL has many moving parts"**
>
> Thank you for the comment. We recently found that, by using a better aggregation of double Q functions, we can completely **remove two hyperparameters of TRL** (quantile $\eta$ and subgoal count), while maintaining (or sometimes achieving even better) performance. Now, TRL has only *two* major hyperparameters that require task-dependent tuning (the BC coefficient $\alpha$ and the distance re-weighting factor $\lambda$; Table 5). This number (2) of per-task hyperparameters is smaller than or equal to those of the closest prior works, COE and SGT (Table 5). We have applied this Q aggregation change to both TRL and all relevant baselines (which ensures a fair comparison), and updated the results and Appendix accordingly. We also note that this update has not affected any conclusions made in this work.
>
> * **Are the SGT-DP and COE baselines fairly tuned?**
>
> Thanks for asking this question. First, we note that these two algorithms were originally evaluated only on tabular or gridworld environments [2, 3]. Indeed, both SGT and COE work well on simple `pointmaze` tasks in our experiments (Table 2), which confirms the validity of our implementation. Moreover, we have extensively tuned the hyperparameters of these methods individually for each task, as shown in Table 5.
>
> Hence, we believe the poor performance of these algorithms is indeed due to their limited scalability to more complex environments. This is somewhat as expected, since the original SGT algorithm directly takes the argmax subgoal over the entire state space (which is prone to (significant) value overestimation), and the COE algorithm implicitly assumes that the state space is differentiable (which is often not the case with contact-rich dynamics in robotics tasks). Indeed, to our knowledge, TRL is the first off-policy divide-and-conquer value learning method that demonstrates strong performance on long-horizon robotic tasks.

---

> > ### Author Response · Authors · 2025-11-20
> >
> > * **BCE vs. MSE**
> >
> > Thanks for the question. First, we would like to note that the use of the BCE loss is a fairly standard trick in the goal-conditioned RL [1, 4]. This is mainly because the BCE loss can better distinguish smaller values (e.g., $\gamma^{1000}$ vs. $\gamma^{2000}$ with $\gamma = 0.99$) than the MSE loss. To experimentally demonstrate this, we conducted an ablation study.
> >
> > | Environment                   | TRL (BCE)   | TRL (MSE)   |
> > |:------------------------------|:------------|:------------|
> > | $\texttt{humanoidmaze-giant}$ | $\mathbf{79} \pm 2$  | $12 \pm 4$  |
> > | $\texttt{puzzle-4x5}$         | $\mathbf{97} \pm 1$  | $81 \pm 8$  |
> > | $\texttt{puzzle-4x6}$         | $\mathbf{51} \pm 5$  | $41 \pm 2$  |
> >
> > The table above compares the performance of TRL with the BCE and MSE losses on the three long-horizon tasks used in Section 5.1. The results suggest that BCE indeed leads to better performance across the board. Finally, we would like to note that **we applied the same BCE trick to all relevant baselines** in the paper (including SAC+BC, TD, TD-n, MC, COE, SGT, and CRL) to ensure a fair comparison.
> >
> > * **"The triangle inequality really needs to be described earlier."**
> >
> > Thanks for the suggestion! Following the suggestion, we have revised the draft to describe the triangle inequality in Section 1.
> >
> > * **"The paper tests on one suite of benchmarks."**
> >
> > While we use OGBench as the main benchmark in this paper, the $13$ different tasks used in this work span highly diverse domains, including robotic navigation, ball dribbling, robotic manipulation, and puzzle solving. We would also like to note that OGBench is currently a standard benchmark in offline goal-conditioned RL, and many previous works in offline (goal-conditioned) RL have evaluated their methods *solely* on this task suite [1, 5, 6, 7, 8]. Given these points, we believe that our empirical evaluation in this paper meets the standard in the literature. However, please feel free to let us know if there are particular missing environments that are necessary to support the claims in this paper.
> >
> > * **Limited to GCRL and deterministic environments**
> >
> > Indeed, the scope of this paper is within goal-conditioned RL, and TRL in its current form is unbiased only in deterministic environments, as in most works in goal-conditioned RL that employ the triangle inequality [2, 3, 9, 10, 11]. However, we expect that TRL could potentially be extended to general reward-based environments, given that any reward-based MDP can be equivalently converted to a (stochastic) goal-conditioned RL problem (e.g., see page 40 of [Neuro-Dynamic Programming](https://web.mit.edu/dimitrib/www/NDP.pdf)), where stochastic environments also satisfy a variant of the triangle inequality [12] that we can use for divide and conquer. We leave this extension as a future research direction (in fact, we're currently actively working on this as a follow-up to TRL).
> >
> > To further clarify that the scope of this paper is within offline goal-conditioned RL, we have revised Section 1 to clearly state that this paper mainly concerns offline goal-conditioned RL (changes in red).
> >
> > ---
> >
> > We would like to thank you again for raising important clarification questions about our work. We believe the additional experiments and clarifications have strengthened the paper. Please let us know if you have any additional concerns or questions. If we have addressed your concerns, we would appreciate it if you could adjust the score accordingly.
> >
> > ---
> >
> > [1] Park et al., Horizon Reduction Makes RL Scalable (2025) \
> > [2] Jurgenson et al., Sub-Goal Trees – a Framework for Goal-Based Reinforcement Learning (2020) \
> > [3] Piekos et al., Efficient Value Propagation with the Compositional Optimality Equation (2023) \
> > [4] Kalashnikov et al., Scalable deep reinforcement learning for vision-based robotic manipulation (2018) \
> > [5] Espinosa-Dice et al., Expressive Value Learning for Scalable Offline Reinforcement Learning (2025) \
> > [6] Agrawalla et al., floq: Training critics via flow-matching for scaling compute in value-based rl (2025) \
> > [7] Ahn et al., Option-aware Temporally Abstracted Value for Offline Goal-Conditioned Reinforcement Learning (2025) \
> > [8] Ke et al., Conservative Offline Goal-Conditioned Implicit V-Learning (2025) \
> > [9] Eysenbach et al., Search on the replay buffer: Bridging planning and reinforcement learning (2019) \
> > [10] Wang et al., Optimal goal-reaching reinforcement learning via quasimetric learning (2023) \
> > [11] Park et al., Foundation policies with Hilbert representations (2024) \
> > [12] Myers et al., Learning temporal distances: Contrastive successor features can provide a metric structure for decision-making (2024)

---

### Official Review · Reviewer_EYPx · 2025-10-31

**Soundness:** 2
**Presentation:** 3
**Contribution:** 3
**Rating:** 6
**Confidence:** 4

**Summary:**

The paper introduces Transitive Reinforcement Learning (TRL), a new divide-and-conquer framework for value learning in offline goal-conditioned reinforcement learning (GCRL). The core idea is to exploit the triangle inequality of temporal distances to decompose long-horizon value estimation into shorter, composable subproblems. They replace the hard maximization over subgoals with soft expectile regression to mitigate overestimation bias and restrict subgoal selection to in-trajectory behavioral subgoals, which further stabilizes learning in the offline setting.

**Strengths:**

1. TRL’s key strength lies in its scalability to long-horizon tasks (up to 4000 steps), where it consistently outperforms or matches the best TD- and MC-based baselines. By reducing the Bellman recursion depth to logarithmic complexity, TRL fundamentally mitigates the bias accumulation problem that plagues TD methods over long trajectories.
2. In contrast to TD-n approaches, TRL attains superior performance without the need for laborious, task-specific tuning of the horizon parameter 𝑛, offering a more robust and parameter-free alternative for long-horizon value learning.

**Weaknesses:**

1. The proposed approach fundamentally depends on deterministic environment dynamics for the triangle inequality assumption to hold. Extending the framework to learn unbiased value functions in stochastic environments remains an important and open avenue for future research.

2. As shown in the ablation study, the results are highly sensitive to subgoal selection, which may introduce additional instability when applied to stochastic or noisy environments.

**Questions:**

1. The ablation study indicates that restricting subgoals to behavioral in-trajectory states is essential for achieving stable performance, whereas using random subgoals leads to a substantial degradation. This sensitivity suggests a strong dependence on subgoal quality and spatial distribution. Would the authors consider exploring an adaptive or learned subgoal sampling mechanism that dynamically selects informative intermediate goals during training?

2. It would be valuable to understand how TRL performs when extended beyond goal-conditioned reinforcement learning to more general reward-based RL tasks. Do the authors expect the divide-and-conquer principle to remain effective in such settings, and have any preliminary experiments been conducted in that direction?

---

> ### Author Response · Authors · 2025-11-20
>
> Thank you for the detailed review and constructive feedback on this work. We especially appreciate the questions regarding the applicability of TRL beyond GCRL and deterministic environments and alternative goal sampling mechanisms. Please find our response below.
>
> * **Limited to GCRL and deterministic environments**
>
> Indeed, the scope of this paper is within goal-conditioned RL, and TRL in its current form is unbiased only in deterministic environments, as in most works in goal-conditioned RL that employ the triangle inequality [1, 2, 3, 6, 7].
>
> Regarding whether TRL can be extended to standard (non-goal-conditioned) RL, we would first like to note that any reward-based MDP can be equivalently converted to a (stochastic) goal-conditioned RL problem (e.g., see page 40 of [Neuro-Dynamic Programming](https://web.mit.edu/dimitrib/www/NDP.pdf)). Although the converted environment has stochastic dynamics, there exists a stochastic variant of the triangle inequality [4], so we may still potentially apply divide-and-conquer to develop a TRL-like algorithm.
>
> Hence, we believe TRL has the potential to be extended to both stochastic and reward-based (standard) RL tasks. However, this would require non-trivial modifications to the algorithm, and thus we believe it is better done as separate future work. We leave this extension as a future research direction (in fact, we're currently actively working on this as a follow-up to TRL).
>
> To further clarify that the scope of this paper is offline goal-conditioned RL, we have revised Section 1 to clearly state that this paper mainly concerns offline goal-conditioned RL (changes in red).
>
> * **"As shown in the ablation study, the results are highly sensitive to subgoal selection, which may introduce additional instability when applied to stochastic or noisy environments." / other subgoal selection mechanisms**
>
> First, we would like to note that the use of our in-trajectory subgoal selection mechanism leads to substantially better performance than random subgoals **across the board** (as shown in our ablation study in Figure 6), which suggests that it is simply a beneficial design choice, rather than a sensitive hyperparameter that requires tuning. Indeed, this is one of the main contributions of this work, as described in Section 4.2.
>
> While one might be concerned that this in-trajectory subgoal selection doesn't explicitly stitch different trajectory segments, we experimentally showed that behavioral subgoals are still highly effective even when the dataset consists of *uniformly random* atomic motions and thus does not necessarily have high subgoal quality. In fact, all OGBench manipulation datasets used in our experiments are collected in this way, and TRL exhibits strong (and often the best) performance on them. This is potentially because stitching in offline GCRL is often performed *implicitly* in practice (i.e., without always requiring explicit stitching), via representation learning and/or scaling [5].
>
> That said, we believe that there may be even better subgoal selection mechanisms beyond in-trajectory subgoals. While we haven't explored such mechanisms in this work (e.g., a mixture of random subgoals and in-trajectory subgoals, learned subgoal generators, etc.), we believe it is an important subject for future studies in divide-and-conquer RL, and we leave it as future work.
>
> ---
>
> We would like to thank you again for raising important questions about potential extensions to our work. Please let us know if you have any additional concerns or questions. If we have fully addressed your concerns, we would appreciate it if you could adjust the score accordingly.
>
> ---
>
> [1] Eysenbach et al., Search on the replay buffer: Bridging planning and reinforcement learning (2019) \
> [2] Wang et al., Optimal goal-reaching reinforcement learning via quasimetric learning (2023) \
> [3] Park et al., Foundation policies with Hilbert representations (2024) \
> [4] Myers et al., Learning temporal distances: Contrastive successor features can provide a metric structure for decision-making (2024) \
> [5] Bortkiewicz et al., Is Temporal Difference Learning the Gold Standard for Stitching in RL? (2025) \
> [6] Jurgenson et al., Sub-Goal Trees – a Framework for Goal-Based Reinforcement Learning (2020) \
> [7] Piekos et al., Efficient Value Propagation with the Compositional Optimality Equation (2023)

---

### Official Review · Reviewer_7UcZ · 2025-10-31

**Soundness:** 3
**Presentation:** 3
**Contribution:** 3
**Rating:** 6
**Confidence:** 4

**Summary:**

This paper presents a novel goal-conditioned RL method with D&C updates. The discussion is restricted to discrete, deterministic RL problems, but I still find the work interesting.

**Strengths:**

1. The paper is well written and easy to follow.
2. The proposed idea is novel and interesting. I agree with the authors that this paper is a first step towards a promising direction.

**Weaknesses:**

1. The authors restrict the discussion to discrete, deterministic environments with trajectory data of equal lengths, although they claim that their proposal can be extended to continuous, stochastic environments and various-length trajectories. I suggest the authors actually do such extensions and present the extended version.
2. The tasks used in the experiments are synthetic without clear real-life purposes.
3. Appendix A is unnecessary. It's just homework-level math.

**Questions:**

1. It is mentioned that “in practice, different trajectories can have different lengths.” I wonder about the technical details of handling trajectories of various lengths.
2. Why is Eq. 7 a multiplication?

---

> ### Author Response · Authors · 2025-11-20
>
> Thank you for the detailed review and constructive feedback on this work. Following the suggestion, we have additionally evaluated TRL on continuous, variable-length datasets. We have also addressed the clarification questions raised in the review. Please find our response below.
>
> * **"The authors restrict the discussion to discrete, deterministic environments with trajectory data of equal lengths"**
>
> Thanks for asking this question. First, we would like to clarify that we assume the state space is discrete and different trajectories have the same length **only for notational simplicity**: TRL can directly be applied to continuous environments and handle variable-length trajectories *without any modifications*.
> Indeed, all environments in our experiments have continuous state and action spaces. While the OGBench datasets happen to consist of trajectories of the same length, there are no issues with running TRL on variable-length datasets. To empirically support this claim, we randomly split the OGBench dataset's trajectories to have different lengths and evaluated TRL in this setting.
>
> | Environment                   | TD         | MC         | TRL        | TRL (variable-length)   |
> |:------------------------------|:-----------|:-----------|:-----------|:------------------------|
> | $\texttt{humanoidmaze-giant}$ | $3 \pm 2$  | $79 \pm 4$ | $79 \pm 2$ | $83 \pm 2$              |
> | $\texttt{puzzle-4x5}$         | $19 \pm 1$ | $47 \pm 7$ | $97 \pm 1$ | $97 \pm 1$              |
>
> The table above compares TRL's performance on the original (same-length) datasets and randomly split variable-length datasets on two long-horizon tasks (we also show the performance of two baselines (TD and MC) for reference). The results suggest that there is no issue for TRL in handling variable-length datasets in practice. To prevent potential confusion about this point, we have revised our paper to clarify that these assumptions are made only for notational simplicity (changes in red).
>
> However, TRL does require the environment dynamics to be deterministic in order to be unbiased. In other words, the deterministic dynamics assumption is not just for notational simplicity, unlike the other two assumptions (discrete spaces and same-length trajectories). We note that this assumption is commonly employed in goal-conditioned RL methods based on the triangle inequality [1, 2, 3, 10, 11]. We acknowledged this point in Sections 3 and explicitly mentioned it as a limitation in Section 6 of the original paper. That said, we believe TRL could potentially be extended to stochastic environments by using a *stochastic triangle inequality* proposed by [4], which we leave for future work.
>
>
> * **"The tasks used in the experiments are synthetic without clear real-life purposes."**
>
> In this work, we employ standard offline goal-conditioned RL benchmark tasks from OGBench, as in many other works in offline (goal-conditioned) RL [5, 6, 7]. While these are simulated tasks, they cover a variety of real-world challenges (long-horizon control, non-Markovian trajectories, temporally correlated actions, etc.) across diverse domains, including robotic manipulation and navigation. Moreover, several works have shown that the conclusions drawn from these tasks could often be applied to real-world tasks [8, 9]. Although evaluating TRL on actual real-world robotic tasks is outside the scope of this work, we believe that it is an exciting future research direction to scale up TRL to even more complex, real-world tasks.
>
> * **"Appendix A is unnecessary"**
>
> The purpose of Appendix A is to show that random splitting of a trajectory (as in the practical TRL algorithm) also leads to $O(\log T)$ average maximum Bellman recursions. We believe this section is of interest to some readers because it partly supports the main motivation presented in Section 1. We would also like to note that the proof is not trivial -- the non-trivial part is to figure out the precise constant factor ($4/3$) to enable the induction in Theorem A.2, where the constant $4/3$ is not obvious from the problem description in L704-L715.
>
> * **"Why is Eq. 7 a multiplication?"**
>
> Thanks for the clarification question! Since $V^\*$ satisfies $V^\*(s, g) = \gamma^{d^\*(s, g)}$, the *additive* triangle inequality in $d$ (i.e., $d^\*(s, g) \leq d^\*(s, w) + d^\*(w, g)$) translates to the *multiplicative* triangle inequality in $V$ (i.e., $V^\*(s, g) \geq V^\*(s, w) V^\*(w, g)$). We have further clarified this point in the updated draft (changes in red).
>
>
> ---
>
> We would like to thank you again for raising important questions about our work. We believe the additional results and clarifications have strengthened the paper. Please let us know if you have any additional concerns or questions. If we have fully addressed your concerns, we would appreciate it if you could adjust the score accordingly.

---

> > ### Author Response · Authors · 2025-11-20
> >
> > [1] Eysenbach et al., Search on the replay buffer: Bridging planning and reinforcement learning (2019) \
> > [2] Wang et al., Optimal goal-reaching reinforcement learning via quasimetric learning (2023) \
> > [3] Park et al., Foundation policies with Hilbert representations (2024) \
> > [4] Myers et al., Learning temporal distances: Contrastive successor features can provide a metric structure for decision-making (2024) \
> > [5] Li et al., Reinforcement Learning with Action Chunking (2025) \
> > [6] Espinosa-Dice et al., Expressive Value Learning for Scalable Offline Reinforcement Learning (2025) \
> > [7] Ke et al., Conservative Offline Goal-Conditioned Implicit V-Learning (2025) \
> > [8] Wagenmaker et al., Steering Your Diffusion Policy with Latent Space Reinforcement Learning (2025) \
> > [9] Zheng et al., Multistep Quasimetric Learning for Scalable Goal-conditioned Reinforcement Learning (2025) \
> > [10] Jurgenson et al., Sub-Goal Trees – a Framework for Goal-Based Reinforcement Learning (2020) \
> > [11] Piekos et al., Efficient Value Propagation with the Compositional Optimality Equation (2023)

---

### Meta-Review · Area_Chair_TMFA · 2026-01-08

**Summary:**

The paper proposes Transitive Reinforcement Learning (TRL), a divide-and-conquer approach, for value learning specifically for offline goal-conditioned reinforcement learning (GCRL). The main idea is to exploit the triangle inequality of temporal distances to decompose long-horizon value estimation into shorter, composable segments. To mitigate overestimation bias, TRL replaces the hard maximization over subgoals with soft expectile regression and restricts selected subgoals to in-trajectory behavioral subgoals to further stabilize offline learning.


The main strengths of this work:
- An insightful connection between triangle inequality and value learning in goal-conditioned RL.
- Reviewers all appreciate the importance of the value function learning in long-horizon problems and the novelty of the proposed divide-and-conquer approach as well as several key enhancements in implementation.
- Impressive and extensive experimental results on goal-reaching tasks in OGBench, especially for long-horizon problems.

The reviewers also raised the following main concerns:

(1) TRL is heavily constrained to deterministic, discrete, goal-conditioned settings (Reviewers 7UcZ, EYPx, KdC6, 6EwX):

(i) All the reviewers noted that the framework fundamentally relies on deterministic dynamics (e.g., for the triangle inequality to hold) and is restricted to offline goal-conditioned RL with equal-length trajectories. While the paper claims extensibility to stochastic, continuous, or reward-based settings, these extensions are not theoretically supported nor empirically demonstrated. (ii) Moreover, the paper’s framing shall be revised to emphasize the scope of TRL (i.e., for offline goal-conditioned RL).

(2) Sensitivity to subgoal selection (Reviewers EYPx, 6EwX):

Reviewers highlighted that performance depends critically on the chosen in-trajectory subgoals, with random or poorly distributed subgoals leading to performance degradation.

(3) Regarding the algorithm design and implementation (Reviewer KdC6):

One reviewer noted that TRL relies on oracle representations and has many moving parts with many hyperparameters to tune in practice.

(4) Regarding experimental evaluation and baselines (Reviewers 7UcZ, 6EwX, KdC6):

(i) Reviewers noted the use of largely synthetic tasks in OGBench without clear real-world relevance and the absence of stronger or more recent baselines in offline GCRL (e.g., HIQL and WGCSL). (ii) One reviewer also asked about whether some of the baselines (SGT-DP and COE) are properly tuned. (iii) More empirical justification needed to verify the performance gain from divide-and-conquer value learning.

**Reviewer Concerns:**

After the rebuttal, all of the above concerns have been either fully addressed or alleviated. Specifically:

Regarding (1), during the rebuttal, the authors clarified that TRL can be directly applied to continuous environments and handle variable-length trajectories without any modifications. This is supported by some preliminary experiments on two goal-oriented tasks.
However, the authors also acknowledged that the assumption of deterministic transitions is necessary and also clarified the scope in the revised paper accordingly.

As for (2), the concern is largely alleviated given that the use of in-trajectory subgoals is conceptually reasonable and supported by the experimental evidence on various GCRL tasks.

The rebuttal addressed (3) and (4) through additional experiments, including an ablation on oracle representation and comparison to HIQL and WGCSL, and clarified the hyperparameter choices in the appendix. While this work focuses on OGBench for the experimental evaluation, I find it acceptable as the results already demonstrate the promise of TRL in long-horizon GCRL.

**Reviewer Scores:**

Initially the paper received mixed reviews: 7UcZ: 6 / EYPx: 6 / KdC6: 4 / 6EwX: 4.

During the discussions, Reviewer 6EwX noted that the concerns were mostly addressed, provided some additional suggestions about the presentation, and raised the score to 6 accordingly.

Upon a careful read of the other three reviews and the rebuttal, the major concerns mentioned above appear mostly answered despite the limited scope of this paper (deterministic offline GCRL).

Nevertheless, this work introduces a new and interesting direction for long-horizon GCRL with promising results and likely can spark more future exploration. Overall I find the strengths of this paper to outweigh the remaining weaknesses and recommend acceptance for this paper.

---

### Decision · Program_Chairs · 2026-01-26

Accept (Poster)